

# MoMuCAMS: A new modular platform for boundary layer aerosol and trace gas vertical measurements in extreme environments

Roman Pohorsky[1], Andrea Baccarini[1,2], Julie Tolu[3,4], Lenny H.E. Winkel[3,4], Julia Schmale[1]

[1]Extreme Environments Research Laboratory, Ecole Polytechnique Fédérale de Lausanne, Sion, 1950, Switzerland
[2]Now at: Laboratory for Atmospheric Processes and their Impact, Ecole Polytechnique Fédérale de Lausanne, Lausanne, 1015, Switzerland
[3]Eawag, Swiss Federal Institute of Aquatic Science and Technology, Department of Water Resources and Drinking Water (W+T), Dübendorf, 8600, Switzerland
[4]ETH Zurich, Swiss Federal Institute of Technology, Department of Environment Systems Sciences (D-USYS), Institute of Biogeochemistry and Pollutant Dynamics (IBP), Group of Inorganic Environmental Geochemistry, Universitätstrasse 16, 8092 Zurich, Switzerland

*Correspondance to*: Roman Pohorsky (roman.pohorsky@epfl.ch) and Julia Schmale (julia.schmale@epfl.ch)

## Abstract

The Modular Multiplatform Compatible Air Measurement System (MoMuCAMS) is a newly developed in situ aerosol and trace gas measurement payload for lower atmospheric vertical profiling in extreme environments. MoMuCAMS is a multiplatform compatible system, primarily designed to be attached to a helikite, a rugged tethered balloon type that is suitable for operations in cold and windy conditions. The system addresses the need for detailed vertical observations of atmospheric composition in the boundary layer and lower free-troposphere, especially in polar and alpine regions. These regions are known
to frequently experience strong temperature inversions, preventing vertical mixing of aerosols and trace gases, and therefore reducing the representativeness of ground-based measurements for the vertical column, causing a large informational gap.

The MoMuCAMS encompasses a box that houses instrumentation, a board computer to stream data to the ground for inflight decisions, and a power distribution system. The enclosure has an internal volume of roughly 100 L and can accommodate various combinations of instruments within its 20 kg weight limit. This flexibility represents a unique feature, allowing the
simultaneous study of multiple aerosol properties (number concentration, size distribution, cluster ions, optical properties, chemical composition and morphology), as well as trace gases (e.g. CO, $CO_2$, $O_3$, $N_2O$) and meteorological variables (e.g., wind speed and direction, temperature, relative humidity, pressure) . To the authors' knowledge, it is the first tethered balloon based system equipped with instrumentation providing a full size distribution for aerosol particles starting from 8 nm, which is vital to understanding atmospheric processes of aerosols and their climate impacts through interaction with direct radiation
and clouds.

MoMuCAMS has been deployed during two field campaigns in Swiss Alpine valleys in winter and fall 2021. It has been further deployed in Fairbanks, Alaska (USA) in January-February 2022, as part of the ALPACA (Alaskan Layered Pollution



and Chemical Analysis) campaign and in Pallas, Finland, in September-October 2022, as part of the PaCE2022 (Pallas Cloud Experiment) study. The system flew successfully at temperatures of -36° C, in wind speeds above 15 m s$^{-1}$ and in clouds.

Here we present a full characterization of the specifically developed inlet system and novel, hitherto not yet characterized, instruments, most notably a miniaturized scanning electrical mobility spectrometer and a near-infrared carbon monoxide monitor. Three cases from one of the Swiss Alpine studies are presented to illustrate the capability of MoMuCAMS to perform high-resolution measurements with different instrumental setups. We show two case studies with surface-based inversions in the morning that allowed for observation of aerosol and trace gas dynamics in evolving boundary layer conditions. The vertical

structure of the boundary layer featured in both cases a surface layer (SL) with a top between 50 and 70 m above ground level, dominated by traffic emissions leading to particle number concentrations up to seven times higher than in the residual layer above. Following sunrise, turbulent mixing led to rapid development of a mixed boundary layer and dilution of the SL within one to two hours. The third case study illustrates the capability of the system to perform aerosol sampling at a chosen altitude over several hours, long enough in low aerosol concentrations environments to perform chemical analyses. Trace elements

were analyzed using inductively coupled plasma tandem mass spectrometry. The samples were also analyzed under a scanning electron microscope with energy dispersive x-ray and a transmission electron microscope to gain additional insights into their morphology and chemical composition. Such analyses are suitable to gain deeper insights into particles' origins, and their physical and chemical transformation in the atmosphere.

Overall, MoMuCAMS is an easily deployable tethered balloon payload with high flexibility, able to cope with the rough

conditions of extreme environments. Compared to uncrewed aerial vehicles (drones) it allows to observe aerosol processes in detail over multiple hours providing insights on their vertical distribution and processes, e.g. in clouds, that were difficult to obtain beforehand.

**Introduction**

One of the key challenges in aerosol science is understanding the large heterogeneity of particles in space and time. A particular

gap exists in the knowledge of the vertical distribution and properties of aerosols since most detailed measurements are conducted at the surface. However, the vertical distribution of particles matters, in particular for their climatic effects (Carslaw, 2022). Aerosols interact directly with solar radiation by scattering and absorption, and indirectly as they influence the formation and properties of clouds (Boucher et al., 2013; Haywood and Boucher, 2000; Seinfeld and Pandis, 2016). In particular, subsets of particles, called cloud condensation nuclei (CCN) and ice nucleating particles (INP), can form liquid cloud droplets and ice

crystals, respectively. For particles to affect clouds, they need to be transported to the height where clouds form. For the direct radiation interactions, the vertical location of absorbing aerosols matters specifically (Samset et al., 2013), because the absorbed energy causes local heating which stabilizes the temperature profile in the atmosphere with a variety of consequences such as cloud burn-off. Knowing the aerosols' vertical distribution can improve our estimates of aerosol radiative forcing, which is still the largest single contributor to uncertainty in anthropogenic radiative forcing (IPCC, 2021).



Understanding the vertical distribution becomes particularly important in environments, where the atmospheric boundary layer (ABL) is strongly stratified, such as in polar and mountainous regions (Chazette et al., 2005; Graversen et al., 2008; Harnisch et al., 2009; Persson et al., 2002). The stratification leads to the layering of aerosols and reduced exchange processes, meaning that ground-based measurements are often not representative of cloud-level aerosol (Brock et al., 2011; Creamean et al., 2021; Jacob et al., 2010; McNaughton et al., 2011). Because the ABL represents an exchange interface between the surface and the

free troposphere (FT), it is highly relevant to study the different physical, chemical and dynamical processes that aerosol particles undergo in this lower part of the atmosphere (Jin et al., 2021; Kowol-Santen et al., 2001). Better constraining these processes will help determine to what extent aerosol particles will or will not be present at higher altitudes but also how particles will potentially mix down to the surface. The lack of observations strongly inhibits us from constraining numerical models, which do not perform well in representing the vertical structure of aerosol properties (Koffi et al., 2016; Sand et al.,

2017). However, for assessing the direct and indirect radiative impact of aerosols knowing their vertical distribution is vital. Remote sensing measurements from satellites or ground-based stations offer opportunities for large scale and/or continuous coverage. Nevertheless, remote sensing methods lack detailed information on particle composition and microphysics, and the temporal and spatial resolution is often too coarse for a detailed characterization of aerosol vertical processes (Gui et al., 2016; Mei et al., 2013). Furthermore, retrieval algorithms need validation and this can only be done with in situ measurements.

Shortcomings are particularly large in polar and mountain regions, where space-born aerosol-focused remote sensing (e.g., Cloud-Aerosol Lidar and Infrared Pathfinder Satellite Observation, CALIPSO) provides nearly no data north of 82°N, signals become attenuated under thick clouds, sensors are challenged by surface brightness, and aerosol concentrations are often too low (Kim et al., 2017; Mei et al., 2013; Thorsen & Fu, 2015). Ground-based remote sensing is limited in vertical resolution, because retrievals do not start at the surface but further aloft, which is a key problem in regions with very shallow inversions.

In situ measurements from aircraft have provided valuable information (e.g. Pratt & Prather, 2010; Schmale et al., 2010, 2011), but they remain logistically challenging, expensive, and sometimes cannot be carried out in complex and foggy terrain. Measurements at high speed can also cause flow-induced issues (Spanu et al., 2020) and do not allow for the observation of processes that unfold over minutes to hours such as mixing of air layers and cloud formation. Moreover, typically aircraft do not fly within the first hundreds of meters above the ground, missing therefore valuable information.

UAVs (uncrewed aerial vehicles) and tethered balloons are two effective alternative types of platforms for vertical in situ measurements of aerosol properties. UAVs offer advantages in terms of spatial coverage and flight pattern flexibility but are often limited in their lifting capacity and available space and weight for the payload. Tethered balloons represent a valuable alternative with better lifting capacities, extended flight duration (only limited by available power for instruments) and the ability to collect very high spatial resolution vertical profiles. Recently, there have been important developments in both types

of systems, UAVs and tethered balloons (e.g. Bates et al., 2013; Creamean et al., 2018; Ferrero et al., 2016; Mazzola et al., 2016; Pilz et al., 2022; Porter et al., 2020). Focusing on tethered balloons, the HOVERCAT (Creamean et al., 2018) and the SHARK (Porter et al., 2020) have been developed to measure mainly INP. The AGAP (Mazzola et al., 2016) and the CAMP (Pilz et al., 2022) combine measurements of aerosol optical properties, aerosol number concentration and provide some



information about particle size distribution, mainly based on optical particle counters. In addition, the AGAP also measures
ozone (O$_3$) mixing ratios. This list is non-exhaustive and in addition, other tethered balloon systems have also been developed
to study cloud microphysics and atmospheric turbulences (e.g. Canut et al., 2016; Pasquier et al., 2022). The systems referenced
above have typically been designed for specific targets and have therefore limited freedom in instrumental setup modification.
Here we present MoMuCAMS (Modular Multiplatform Compatible Air Measurement System), a new system for vertical
measurements in the lower atmosphere that has been specifically designed with the aim to remain modular. It combines
instruments for aerosol properties, trace gas and meteorological measurements, which can be combined in different
configurations from one flight to another to provide a more comprehensive view on the various processes in the lower
atmosphere. Additionally, to the authors' best knowledge, MoMuCAMS is the first tethered balloon-based system providing
a full particle size distribution (PSD) from 8 to 3000 nm. Being able to identify the number concentrations and properties of
particles in the CCN size range (> 100 nm) and in the optically most important size range (~500 – 1000 nm, where the aerosol
scattering efficiency is highest) (Seinfeld and Pandis, 2016), is critical to reduce uncertainties in anthropogenic radiative
forcing. The system addresses thus the need for measurements in the lower atmosphere in extreme environments with cold and
windy conditions, where there is a particularly large informational gap. MoMuCAMS has been primarily designed to be
attached under a helikite (Allsopp Helikite, UK). Helikites are rugged and offer the advantage of gaining lift and remaining
very stable under windy conditions, while most other tethered balloon systems typically cannot fly under such conditions. Our
system can be operated by only two people and is light enough (<100 kg for the balloon setup, ~60 kg per winch and ~20 kg
for the payload) to be deployed on sea ice.

MoMuCAMS has been tested during two field campaigns in Swiss Alpine valleys in winter and fall 2021. It has been further
deployed in Fairbanks, Alaska in January-February 2022, as part of the ALPACA (Alaskan Layered Pollution and Chemical
Analysis) (Simpson et al., 2019) field campaign and in Pallas, Finland, in September-October 2022, as part of the PaCE2022
(Pallas Cloud Experiment) (Doulgeris et al., 2022) intensive field study.

This manuscript provides a detailed description and characterization of the MoMuCAMS system and various instruments
under Sect. 2 and 3. Three case studies are presented in Sect. 4 to demonstrate the system's capabilities.

## 2 Technical description of payload and tethered balloon

### 2.1 MoMuCAMS payload characteristics

MoMuCAMS is a modular aerosol and trace gas measurement platform designed to be flown under a tethered balloon, while
it can also be operated from other "tethers" (ropes) such as from cranes or alongside towers and tall buildings. The novelty of
this platform lies in its flexibility to accommodate a very large number of combinations of instruments within the weight and
dimension limits. A list of instruments, which MoMuCAMS typically flies, is presented in Table 1. Importantly, MoMuCAMS
is designed to accommodate guest instruments and can easily be adapted for additional instruments.



The payload enclosure is a box with outer dimensions of 80 x 40 x 35 cm and a cone-shaped nose in the front (see Fig. 1). It provides a total inner volume of roughly 100 liters for instruments and batteries, which can be placed on two levels ("shelves") or attached on the outside. The box is made of 20 mm thick extruded polystyrene plates. This material was selected for its low weight, rigidity and thermal insulation properties. Two aluminum T-elements placed at the front and back of the box support the enclosure from underneath and are used to attach it to the balloon. This system guarantees the stability of the payload in

the air. The box weighs (including the power distribution system and aluminum reinforcements) 3.2 kg. The instruments are powered by lithium-polymer (LiPo) batteries. The system is equipped with two 20 W resistive heaters connected to a thermostat to ensure the inner environment of the box remains above 0° C.

A custom-made data logging and communication system has been designed for MoMuCAMS. A Teensy 3.6 microcontroller programmed with Arduino IDE controls the different tasks. The microcontroller saves data from onboard sensors measuring

internal temperature, barometric pressure, external and sampled air temperature and relative humidity, battery state of charge, particle number concentration from an optical particle counter and $CO_2$ mixing ratio. Data are also simultaneously streamed to the ground through an Xbee 3.0 radio module. Figure 2 shows a schematic sketch of the inner design.

A subset of the data is visualized live on a graphical interface, which helps for decision-making and sampling strategy adaptation during flights. Additionally, the operator can use the graphical interface to send commands to the MoMuCAMS

microcontroller to control various instruments remotely.

## 2.2 Helikite

A 45-cubic meter Desert Star helikite from *Allsopp Helikites ltd* is used to lift the payload. The balloon consists of an outer shell and an inner membrane, which contains the helium. The payload and helikite can be operated by two people only. The

maximum line extension is 800 m. A helikite combines lifting capacity from the helium and from a kite, providing higher lift and good stability in windy conditions. The lifting capability of the helikite depends on the take-off altitude, i.e. atmospheric pressure, and wind speed, and is generally sufficient to lift a payload between 12 and 20 kg. The helikite has been selected for its rugged characteristics, which allow for deployments in the harsh environmental conditions of polar and mountain regions. The helikite has successfully flown at wind speeds up to 15 m s$^{-1}$, in temperatures down to −36° C, and in clouds. Note that

when the air reaches very low temperatures (we estimate that -20° C represents a critical threshold), small punctures form in the balloon's inner membrane, which will consequently lead to helium losses over time and reduced operation time (the inner membrane has to be repaired or replaced).

## 3 Payload instrument characterization

In this section, we provide a detailed characterization of the inlet system (Sect. 3.1), and instruments used on MoMuCAMS

which have not already been described in previous publications. In particular, we present the aMCPC (Sect. 3.2, see Table 1



for abbreviation), mSEMS (Sect. 3.3) and Mira Pico gas analyzer (Sect. 3.5). The POPS was described already by Gao et al. (2016) and Mei et al. (2020); nonetheless, we present here a characterization of our POPS (Sect. 3.4) because it constitutes a reference instrument on the MoMuCAMS. Additionally, filter-based sample collection and associated analytical techniques for chemical composition and electron microscopy are described in Sect. 3.6 and 3.7, respectively. Performance of a

meteorological sensor (SmartTether) is presented in Sect. 3.8. The reader is referred to Pikridas et al. (2019) and Pilz et al. (2022) for a description of the STAP (model 9406, Brechtel Manufacturing Inc.).

## 3.1. Inlet sampling efficiency and transmission losses

The inlet system is composed of a horizontal 30-cm long 3/8" stainless steel tube at the front of the box. Because the tethered

balloon orients with the wind, the inlet is always facing into the wind direction. The tip of the inlet has a 30° downward bend to prevent water droplets from entering. A flexible thermofoil around the inlet heats the sample flow to reduce relative humidity to < 40 %, which corresponds to Global Atmosphere Watch standards (World Meteorological Organization, 2016), and prevent ice formation when sampling in cold environments. The inlet heating is controlled by a miniaturized thermostat (CT325, Minco) and set to be always above 0° C or ~10° C higher if ambient temperature is positive. Sample air temperature and

relative humidity are monitored by a sensor (SHT80, Sensirion). The sensor is placed inside the sampling line in parallel to the instruments to avoid particle losses. The sampled air is split into 1/4" branches and conductive black silicon tubing distributes the sampled air to the different instruments. Additionally, gas sensors such as the ozone monitor, and the stage impactor have their own inlet made of Teflon and Tygon, respectively. The carbon dioxide sensor is installed on the outside of the box and measures air flowing through passively.

The overall sampling performance of the main inlet has been characterized both experimentally and with the Particle Loss Calculator (PLC) (von der Weiden et al., 2009). Sampling efficiency (see Fig. 3) has been computed for wind speeds between 0 and 10 m s$^{-1}$, representative of most operating conditions, and a total sampling flow of 1.72 lpm, which is representative of a typical instrumental setup installed on MoMuCAMS. The flowrate may slightly vary from one setup to another. Results from the PLC indicate that oversampling, due to super-kinetic conditions, becomes important only for larger particles (> 2 μm) at

higher wind speeds.

Transmission losses in the inlet have been experimentally tested with polystyrene latex spheres (PSL) of different diameters ($D_P$). The nebulized PSLs were first dried through a silica gel column (similar to the TSI 3062 type) and the size selection was refined through a Differential Mobility Analyzer (DMA). A reference condensation particle counter (CPC) measured the particle number concentration after the DMA, while two CPCs were placed after the inlet. To represent the different tubing

lengths inside the payload, one CPC was placed behind a short piece of black tubing (10 cm) and one was placed behind a longer piece (45 cm). The total flow through the main inlet was 1.72 lpm. Before the experiment, all CPCs were connected in parallel for direct comparison. Results from the CPC intercomparison are presented in Sect. 3.2.



Figure 3b shows the results of the inlet transmission test (colored dots with error bars) for six different PSL diameters and from the PLC for particles ranging from 8 to 3000 nm. Generally, results compare well between the experiment and the PLC

with slightly lower losses for the shorter inlet. Transmission efficiency for particles between 50 and 1000 nm is very close to 100 % while smaller particles suffer from diffusional losses and larger particles from gravitational deposition. However, the losses are typically less than 10 %. Note that concentrations for PSLs with diameters of 510 and 995 nm were around 35 cm$^{-3}$ and 20 cm$^{-3}$, respectively. Therefore, small variation in the absolute concentration of one CPC might have had a large impact on the transmission efficiency calculation. This is probably the reason for the apparently large discrepancy with the short inlet

measurement for 994 nm particles: the ~25 % difference is explained by just four or five particles cm$^{-3}$ absolute difference (see bottom right purple dot on Fig. 3).

**3.2 Advanced Mixing Condensation Particle Counter (aMCPC)**

The compact advanced mixing condensation particle counter (aMCPC model 9403, Brechtel Manufacturing Inc) is used for

total particle concentration measurements from 7 to 2000 nm, and weighs 1.7 kg. Two aMCPCs have been compared against a reference MCPC (MCPC model 1720, Brechtel Manufacturing Inc, 2.7 kg) with PSLs of $D_P$ 150 nm. PSLs were nebulized and dried as described in Sect. 3.1. The two aMCPCs and the reference MCPC were connected in parallel behind the drier. Figure S1 in the supplementary material shows results of the experiment. Both aMCPCs agree well (within 5%) with the reference MCPC.

In addition, both aMCPCs' counting efficiency as a function of particle diameter was tested experimentally. The counting efficiency was calculated by comparing concentrations measured by the aMCPCs and a reference ultrafine CPC (ultrafine CPC3776, TSI). The $d_{50}$ cutoff, defined as the diameter where the counting efficiency reaches 50%, was found to be equal to 5.7 and 6 nm for aMCPC 21 and 22, respectively. The detection efficiency for both aMCPCs reaches a plateau around 7 nm, in agreement with the manufacturer's specifications. Details are presented in the supplementary material (S1).


**3.3 Miniaturized Scanning Electrical Mobility Sizer (mSEMS)**

The miniaturized Scanning Electrical Mobility Sizer (mSEMS model 9404, Brechtel Manufacturing Inc) is a compact particle size spectrometer providing PSD based on the mobility diameter for particles between 8 and 300 nm. The instrument is composed of a soft X-ray aerosol charge neutralizer (Soft X-ray Charger XRC-05, HTC), a miniaturized DMA (Differential

Mobility Analyzer) column and an aMCPC with a total weight of 4.4 kg. The design of the DMA has been optimized to minimize the high voltage required for particle selection and therefore reduces problems of arching at higher relative humidity or lower pressure. The small internal volumes of the DMA and inlet tubing, and the fast aMCPC time response facilitate rapid scanning due to minimal smearing/mixing volumes inside the instrument.



The performance of the mSEMS was tested with PSL. The mobility diameter ($D_{mob}$) was obtained by fitting a lognormal
distribution to the measured PSD and taking the peak value (mean). The uncertainty of the mean diameter was defined as one
standard deviation of the fitted distribution. Results of the experiments are presented in Fig. 4a. Maximum deviation of
measured $D_{mob}$ are 8% and 3.1% for 51 and 70 nm PSL, and below 1% for 150 and 240 nm PSL.

In addition, particle transmission through the neutralizer and DMA has been tested for different PSL sizes. For the experiment,
PSL particles were nebulized and size selected with a first DMA. A standalone aMCPC was connected in parallel to the
mSEMS after the first DMA. Transmission through the mSEMS (neutralizer + DMA) was calculated by comparing the particle
number concentration measured by the two aMCPCs. Results are presented in Fig. 4b. A sinusoidal function (Eq. 1):

$$f(Dp) = \frac{A}{1+\exp(-B*(Dp-x_0))} \tag{1}$$

with the following fit results A = 1.00, B = 0.14 and $x_0$ = 13.46 , where $x_0$ is the 50% transmission point that was used to fit
the experimental transmission results. Based on the measured losses below 30 nm, a correction is applied to the mSEMS data
obtained in the field using Eq. (1). Figure 5 shows results of 10-minute averaged integrated particle number concentrations
from the mSEMS against a standalone aMCPC measuring in parallel. Data was collected from a ground measurement station
in Brigerbad, Switzerland between October 8 and October 11, 2021 (see Sect. 4 for campaign details). Figure 5a shows results
for the original mSEMS data and Fig. 5b shows results after data correction. The color scale indicates the number concentration
($N_{8-30}$) of particles with $D_{mob}$ between 8 and 30 nm to highlight the higher discrepancies between the mSEMS and the aMCPC
when the number of ultrafine particles increases. Dots indicating higher $N_{8-30}$ are typically further away from the 1:1 line (Fig.
5a), confirming an underestimation of total number concentration because of ultrafine particle losses through the neutralizer
and DMA. By applying the empirical transmission loss correction function, the slope of the linear regression increases from
0.61 to 0.79 and the scatter in the data is reduced ($R^2$ increases from 0.94 to 0.99, Fig. 5b). The remaining underestimation of
the particle concentration can be explained by the narrower size range counted by the mSEMS (8 to 280 nm) compared to the
aMCPC (7 to 2000 nm). These measurements show that ultrafine particle losses in the mSEMS are non-negligible and a
correction factor should be applied to improve measurement accuracy.

During flights, the instrument operates with a 0.36 lpm sample flow and 2.5 lpm sheath flow. We typically select a size range
from 8 to 280 nm with 60 bins and a scan time of 1 minute (up scan). Given the lower time resolution of the mSEMS compared
to other instruments onboard MoMuCAMS, a typical flight strategy consists in a fast ascending profile followed by a stepwise
descent. Distance between each step varies according to the maximum altitude of the profile, desired time of flight and
atmospheric conditions such as temperature inversions or presence of stratified layers. A typical stop at a fixed altitude lasts
at least five minutes to allow a minimum of five full scans for better counting statistics of the measured PSD. An example of
such a flight pattern is presented in Sect. 4.2.



### 3.4 Portable Optical Particle Spectrometer (POPS)

The well-characterized Portable Optical Particle Spectrometer (POPS, Handix Scientific) is used to obtain PSD and number concentrations of particles between 186 and 3300 nm (Gao et al., 2016; Mei et al., 2020). A sizing and counting efficiency
calibration of two POPS (1 for flights [POPS105] and 1 for ground measurements [POPS101]) was performed with PSL. Results are presented in the supplementary material (Fig. S3). POPS105 shows size deviations below 10% for PSLs up to 800 nm while POPS101 shows slightly higher deviations up to 20% for 500 nm particles. Both POPS show higher deviation for 994 nm particles: 34% and 29% for POPS 101 and 105, respectively. The higher deviation for particles around 1 µm can be explained by Mie resonance in this size range and has also been observed by Pilz et al. (2022). We follow therefore their
recommendations by setting the POPS size resolution to 16 bins (instead of a higher number) to reduce sizing artefacts.
Based on counting efficiency tests (see details in the supplementary material, Fig. S4), it appeared that particles with diameters between 142 and 186 (bins 1 to 3) are wrongly detected by the POPS as total particle concentration increases. This phenomenon can be explained by electronic noise from the detector, where fringes on the edge of the Gaussian signal are perceived as smaller particles by the software. It was therefore decided to only consider data for particles larger than 186 nm as the error
induced by the first three bins is too high. Overall, both POPS show very good agreement with the reference CPC with deviation below 10% for the total number concentrations.

### 3.5 Mira Pico CO/$N_2O$/$H_2O$ analyzer

The Pico (Mira Pico CO/$N_2O$, Aeris Technologies) is a compact NDIR-based (non-dispersive infrared) gas analyzer. The
instrument uses middle-infrared laser absorption spectroscopy to measure CO, $N_2O$ dry mole fraction and $H_2O$ with a sub ppb detection limit. Only a few studies have provided information on the performance of the Pico instrument, however only for the methane ($CH_4$) version (Commane et al., 2022; Travis et al., 2020).To the authors' best knowledge, this study is the first report on the CO version.
The instrument is integrated inside a small Pelican case (L30 x W20 x H9 cm) and weighs 2.7 kg, including a battery with a
6-hour lifetime. The Pico can work in two different modes. The instrument is equipped with two programmable sampling ports. In its differential mode, the system switches between the two sampling ports at a user definable time interval (30 second by default). A catalytic CO-scrubber is placed in front of the first port, providing a zero measurement for each interval, effectively preventing any slow instrument drift. The software automatically removes the baseline (zero measurement) from the actual measurement. In this configuration, the Pico provides measurements at a 1-minute time resolution with a 1-ppb
accuracy (the value is provided by the manufacturer but has not been validated experimentally). In its manual mode, the instrument samples only from one port with a 1-second time resolution. In this configuration, no baseline correction is applied to the measurements, reducing the overall accuracy. To estimate the reduction in accuracy due to unaccounted baseline drifts occurring over a typical flight period, we analyzed zero measurements (i.e., CO scrubber installed in front of sampling port



and Pico operating in manual mode) for 90 minutes. We consider two standard deviations of the zero measurement distribution

as an upper limit estimate of the measurement uncertainty in manual mode; this value is equal to 17 ppb.

For flight operation, the manual mode is preferred to provide the highest time resolution possible. To account for the baseline, the instrument is operated on the ground between flights in its differential mode. Before each flight, the instrument is placed inside the box and brought outside until temperature inside the box has stabilized. The CO-scrubber is removed and the Pico set to manual mode just before take-off. The baseline measurement for the last 3 hours before the flight and 3 hours after the

flight is then averaged and subtracted from the flight measurements. This operation should provide the best estimate for the baseline deduction from the measured values. To identify, whether pressure or temperature changes have any influence on the instrument's baseline, several flights were performed in differential mode. No evident link between payload inner temperature, ambient pressure and baseline variation was found, ultimately showing that baseline variability during and between flights was similar (see Fig. 6b). Figure 6a shows the baseline measurement for a full campaign with color codes indicating whether the

instrument was operated on the ground or in the air. Figure 6b shows in more detail the baseline variability on January 30, before, during and after a flight. Results indicate that the baseline remains stable over the campaign and the flight does not affect the variability.

Note that during measurements, we recommend to save the high time resolution spectral files to control good data fitting or to detect fitting issues. In case of fitting issues, the spectral files can be processed again to correct the data.

So far, no quantitative characterization of the Pico's performance is available. A comparison with a reference instrument or calibration gas should be done for future quantitative assessments of CO with the Pico.

### 3.6 Filter sampling for chemical analyses

In addition to online measurements, the MoMuCAMS system can also be equipped with instruments for offline analysis. Two

instruments are currently used to collect aerosol samples on filters for chemical and microscopic analyses. A more detailed description of the instrumental setup, types of analysis, filter preparation and handling, and analytical procedures is given below.

A high-flow multi-stage cascade impactor (HFI Model 131A, TSI) is used to collect aerosol particles on filters. The multiple nozzle-pattern achieves cut-size selection similarly to the more common Micro-Orifice Uniform-Deposit Impactors (MOUDI).

A nominal sampling flow of 100 lpm is achieved by a radial flow impeller (Radial blower U85HL-024KH-4) used in reverse as a lightweight pump as in Porter et al. (2020). The sampling flow is constantly monitored by a flowmeter installed before the blower (SFM3000, Sensirion). The HFI is equipped with 6 stages with the following cutoffs: 10, 2.5, 1.4, 1.0, 0.44 and 0.25 µm. Samples are collected for the 6 size cutoffs on 75 mm diameter quartz fiber filters (QR-100, 0.38 mm thickness, Advantec MFS Inc.) and then on a 90 mm diameter quartz fiber filters (AQFA, Merck Millipore ltd) to collect all particles

below the lowest cutoff.



For airborne sampling, the number of stages used is usually reduced to three to optimize mass collection on filters, especially if sampling time is reduced because of flight duration restrictions imposed by regulations. A typical sampling strategy consists in bringing the tethered balloon to a desired sampling altitude where it will hover. The blower can be activated and deactivated remotely and the flow can be controlled.

Before sampling, all filters are baked for 6 hours at 550° C in separate aluminium pouches to reduce contaminants in the blanks and directly sealed in plastic zip-bags. We collect regular blanks for each sampling campaign. In particular, we have two types of campaign blanks: regular blanks and field blanks. The former are brought to the field but not taken out from their aluminium pouches (regular blank). The latter are installed in the filter sampler and retrieved shortly after to mimic field operations (field blank). After sampling, loaded filters are retrieved, folded in half and placed back in their respective pouches. Retrieval of

filters is performed, if possible, at temperature conditions similar to sampling conditions to avoid any evaporation of volatile compounds. Filters are then stored at -20° C before analysis.

To quantify element concentrations in collected aerosols, half of the filters are first digested in a mixture of nitric acid (69% $HNO_3$, Suprapur; Roth), hydrogen peroxide (30% $H_2O_2$, for ultratrace analysis; Sigma-Aldrich) and ultrapure water (18.2 MΩ cm; Nanopure DIamond$^{TM}$ system) using an MLS GmbH UltraCLAVE 4 microwave. Elements are then quantified in the

digests using an Agilent 8900 inductively coupled plasma tandem mass spectrometry (ICP-MS/MS) (for a detailed description of the digestion and analysis, the reader is referred to the supplementary material, Sect. S3 and Table S1). The analysis of the blank filters is used to determine detection limits and whether field manipulation affects the background contamination of filters. Hereafter results of the background levels for Cu and Se are presented in Table 2 as a reference for the case study presented in Sect. 4.3. The resulting detection limits are calculated according to IUPAC recommendation (McNaught and

Wilkinson, 1997), i.e., the mean plus three times the standard deviation of obtained blank concentrations. The background levels obtained for other trace elements and resulting detection limits are presented in the supplemental material (Sect. S3, Table S2). Results of regular and field blanks revealed no difference in the levels of trace elements, suggesting that the substrate itself and the digestion step are the largest sources of contaminations.

**3.7 Filter sampling for Electron Microscopy**

An 8-channel filter sampler (FILT Model 9401, Brechtel Manufacturing Inc) is used to collect samples on substrates for electron microscopy analysis. Each channel holds a 13-mm Teflon Swinney filter holder. Polycarbonate filters with 0.4 µm pores (ref. number 321031, Milian Dutscher Group) are used to collect particles for scanning electron microscopy with energy dispersive x-ray analysis (SEM/EDX). Polycarbonate filters offer a smooth surface and are mechanically rugged (Genga et al.,

2018; Willis and Blanchard, 2002), which is ideal for particle observation and prevents deterioration of the substrate during sampling.

For Transmission Electron Microscopy (TEM) analysis, custom-made TEM grid holders were created to fit the standard 13-mm filter holders (see Fig. 7). Additionally, a "jetting" device (Brechtel Manufacturing Inc., USA), placed above the grid,



reduces the inlet diameter and focuses the sampling beam onto the TEM grid. The real particle impaction efficiency has
however not been characterized so far.

The filter sampler can operate between 0.5 and 3 lpm. However, the pump does not sustain a sampling flow above 1.8 lpm
with the additional TEM grid holder. Furthermore, higher sampling flows tend to destroy the grid's carbon membrane.
Therefore, we operated the FILT with a sampling flow of 1.5 lpm. Both the sample flow and the sampling stage can be remotely
controlled from the ground. After filter retrieval, filters are stored at -20° C until analysis. Airborne sampling was first
performed in October 2021, in a Swiss Alpine valley. Examples of collected aerosol particles with SEM/EDX and TEM are
presented in Fig. 8.

For scanning electron microscopy, the analysis is carried out on a Thermo-Scientific Teneo. This machine is equipped with a
Bruker XFlash EDX detector, as well as Everhart-Thornley and Trinity (in-column) electron detectors. Imaging and EDX
spectroscopy are performed using a beam energy of 5 keV. A focused electron probe is scanned over a region of interest to
collect EDX data in the form of spectrum images. For each region of interest, a second EDX map using a beam energy of 15
kV is acquired in case of ambiguity or peaks that overlap. To account for the signal from the sampling substrate, the beam is
first focused on an aerosol free substrate area (red trace in Fig. 8). Before analysis, filters are coated with a 7-nm iridium layer
to avoid charge accumulation at their surface. Two examples of particles collected during airborne filter sampling on
September 28 and October 7, 2021 are shown on Fig. 8. EDX spectra for particle (a) shows traces of N, O, Fe and Si. Particle
(b) shows traces of N, Si, Al and K. Details on sample collection are presented in Sect. 4.3; however, a full analysis of
SEM/EDX results is beyond the scope of this paper, which serves mainly as proof of concept for airborne aerosol sampling
and subsequent microscopy analysis.

For transmission electron microscopy, the analysis is performed on a Thermo Scientific Tecnai Sprit operating at an
accelerating voltage of 120 kV. The images are acquired under bright field imaging conditions, in which only the directly
transmitted beam, selected by the objective aperture, contributes to the image formation. TEM was performed on collected
samples and confirmed that the system could effectively collect aerosol particles for TEM observations. An example of two
particles collected during the September 28 flight is shown in Fig. S5. Similarly to the SEM/EDX example, these results are
mainly presented for illustrative purposes of the system's capabilities for aerosol sampling and analysis and a more detailed
interpretation is beyond the scope of this paper.

**3.8 Meteorological measurements**

Meteorological parameters including temperature (T), relative humidity (RH), barometric pressure (P), wind speed (WS) and
direction (WD) are measured by a lightweight sonde (SmartTether, Anasphere) placed below the payload. The SmartTether is
contained in a compact plastic casing mounted on a carbon fiber arrow-shaped structure. A cup anemometer is placed at the
front of the structure and a dart-like tail helps the sonde orient itself into the wind.

Table 3 summarizes all measurements and the respective resolution, accuracy and operating range as provided by the
manufacturer. During flight, data is streamed to the ground and directly saved on the ground computer. Note that no data is



saved locally and in case of communication loss, data is not saved. Furthermore, it appears that the SmartTether is sensitive to electromagnetic interferences and frequent loss of communication was experienced in some cases.

Two comparisons were performed on the ground between the SmartTether and a weather station equipped with a HygroVUE10
(Campbell Scientific) sensor, using an SHT35 sensing element (Sensirion, CH). The first comparison was performed in Brigerbad, Switzerland on October 14, 2021. The second comparison was done in Fairbanks, Alaska on February 24, 2022. Figure 9 shows the timeseries of T and RH for both experiments. Additionally, bottom panels show the incoming shortwave radiation flux (measured with an Apogee SN-500-SS). Data from the first comparison indicate that the SmartTether sonde is sensitive to solar radiation (Fig. 9a). In fact, the temperature sensor is directly exposed to the outside and no shield is present
to block radiation. Our tests show that solar radiation leads to a temperature discrepancy of up to 4° C between the two shielded and unshielded sensors. This temperature discrepancy has a direct effect on the temperature dependent RH measurements. Unfortunately, it is not trivial to evaluate how much the sensor is affected by radiation during flights because of the constant motion of the SmartTether. Furthermore, wind might also play a role on how the sensor is affected. A solution including two sensors (SHT85, Sensirion) in a shielding tube with active flow is under development in order to correct for the radiation
sensitivity. Data show good agreement for temperature measurements when solar radiation is low as e.g., on October 13, 2021 after 17:45 and on February 24, 2022 (Fig. 9a and b). On February 24, RH values show a discrepancy up to about 4% (Fig. 9d). This discrepancy could be explained by the higher proximity to the ground of the SmartTether compared to the weather station, and higher uncertainties at high RH values. Overall, the SmartTether provides reliable measurements when solar irradiance is low and/or wind speed is sufficiently high. In other cases, measurements can be biased and data should be treated
accordingly.

## 4 Field application

From September 22 to October 14 2021, MoMuCAMS was deployed in a field campaign to study and characterize the vertical distribution of aerosols and trace gases in an Alpine valley in relation to the complex meteorological conditions of mountain regions. 13 flights were performed during the campaign and a total of 88 profiles were collected (ascending and descending
profiles counted separately). In addition to vertical profiling, ground-based measurements were performed to provide a continuous reference on the ground. A trailer with an inlet system was parked 30 meters from the helikite. Instruments from the MoMuCAMS system sampled from the trailer between flights. Additionally, a SEMS measured PSD from 8 to 1100 nm and a weather station (Campbell Scientific) measured meteorological parameters on the ground.

The study site was located in Brigerbad, Switzerland (46.29°N, 7.92°E), in the Rhône valley at an altitude of 653 m a.m.s.l.
At the site, the valley has an east-west orientation and the valley floor is roughly 500 m wide. Heights of the nearest mountains to the north and south were 2900 and 2300 m, respectively. Typical weather patterns exhibited diurnal temperature cycles during the whole period with an average temperature difference of 9.5° C between the 08:00 minimum and 16:00 maximum. For interpretation purposes, time is given in local time, corresponding to Central European Summer Time (CEST or UTC+2).





In response to the radiation and temperature diurnal cycle, katabatic winds typically blew from the east between 22:00 and
09:00 with a mean velocity of 0.9 m s$^{-1}$. The wind typically transitioned to a cross-valley southerly wind around 10:00 and
further developed into a stronger westerly valley wind in the afternoon. The diurnal cycle was also characterized by surface
temperature inversions occurring frequently during clear sky nights. A rapid dissipation of the inversion layer typically
followed after sunrise. This phenomenon was more marked during the second half of the campaign.

Several anthropogenic sources of atmospheric pollutants are located near the site, including industry, roads, private housing
and agricultural fields. Main contributing sectors to PM$_{2.5}$ and BC have been estimated from the EMEP (European Monitoring
and Evaluation Program) Centre on Emission Inventories and Projections gridded emission database (http://www.emep-
emissions.at/emission-data-webdab/). The percentage contributions by sector to annual emissions in 2020 for PM$_{2.5}$ include
stationary combustion (30.9%), industry (23.5%), off-road vehicles (20%) and road transport (17.1%); and for BC stationary
combustion (55.9%), road transport (22.2%) and off-road vehicles (19.3%) as main contributors.

In the following section, we present case studies illustrating new insights on valley-floor boundary layer processes that
MoMuCAMS offers.

### 4.1 Case 1 – Evolution of aerosol and trace gas concentrations during a surface inversion dissipation

Six profiles (3 ascents and 3 descents) were measured on a cloud-free day on October 1$^{st}$, 2021, from 08:50 to 12:30. Table 4
summarizes the instrumental setup for these flights. Figure 10a shows the ground temperature (T), net radiation (NR) and wind
speed (*U*) and direction evolution from 08:00 to 12:45. At 09:30, the sun rose from behind the mountains, which led to a sharp
increase in NR, followed by a surface temperature increase. Winds at the surface remained low during the flights. Weak
easterly katabatic winds were blowing until roughly 09:30 and then gradually developed into a cross-valley wind around 11:00.
Above 50 meters, winds were slightly stronger (between 2 and 4.5 m s$^{-1}$) and their east-northeast orientation remained rather
constant through the flights (Fig. 11b and c). Figure 10b and c show the ground-based measured PSD and integrated total
concentration (black dots), rising from 08:00 and peaking between 09:00 and 09:30, followed by a gradual decrease until noon,
which is consistent with the onset of convective mixing induced by surface warming. Figure 10d shows a time-series of the
balloon altitude. The color of each altitude point indicates the particle number concentration (N$_{>186}$) from the POPS.

Figures 11 and 12 show 4 different vertical profiles illustrating the evolution of the boundary layer. The selected profiles are
indicated by numbers between brackets in Fig. 10d. Colors indicate the starting time of each profile. Figure 11a show a
temperature inversion with a mean gradient of 1.8° C/100m during the first ascent starting at 08:55 (turquoise profile),
indicative of a stable boundary layer (SBL) up to at least 250 m above ground level (AGL). The top of the inversion cannot be
determined as the maximum reached altitude was still within the inversion layer. Figure 12 shows vertical profiles of particle
number concentration and trace gas mixing ratios. The first profile shows a surface layer (SL) up to 50 m with increased
concentrations compared to more elevated layers (>150 m). N$_{7-186}$ and N$_{186}$ concentrations were up to seven and two times
higher than concentrations measured above 150 m, respectively. Ground-based measurements indicate that surface particle



concentrations started increasing around 08:00 (Fig. 10b) with stable particle concentration before 08:00 (not shown here). The increase at the surface is explained by the morning rush hour and reduced mixing volume due to valley walls and SBL, as has been observed previously in similar valley locations (Chazette et al., 2005 or Harnisch et al., 2009).

Between 80 and 125 m AGL, large peaks in the particle concentration and $CO_2$ mixing ratio were measured during the first ascent. These peaks were, however, not present on the following descent after 09:30 (Fig. 12, orange profile). At maximum peak intensity, the concentration of $N_{7-186}$ and $N_{186-3370}$ was about three and four times larger than above 150 m, respectively. Compared to the SL, $N_{7-186}$ was 1.7 times lower at the plume altitude, but $N_{186-3370}$ was two times larger. The $CO_2$ concentration shows an increase of 10% at the peak compared to surface values. CO exhibits only a weak signal at the same altitude. The

exact origin of the plume is not known. The increase in $CO_2$ mixing ratio might suggest that the particles were recently emitted from an anthropogenic source. The different gas and particle ratios between the SL and the plume layer suggest different source contributions to the two layers. Given the altitude of the plume and the stability of the atmosphere, it can be hypothesized that the source was either located at the same altitude or was located at the surface and had higher injection height. The potential source could thus be either located on the valley slope or be a high stack from an industrial facility. It is not possible to say if

the disappearance of the plume after the first flight was caused by the reduced atmospheric stability, which increased the dispersion and mixing of the plume, or by the termination of the emission process. This measurement provides however clear evidence that MoMuCAMS is effective in detecting plumes aloft and can be used to track emissions at higher elevations.

Not accounting for the above-discussed plume, concentrations in particles and gases decreased between 50 and 150 m (Fig. 12). Wind shear arises from the difference between low winds in the decoupled SL and higher wind speed aloft (Fig. 11b and

c). Mechanical turbulence induced by the shear would explain a slow diffusion of the different SL tracers into the adjacent layer. The diffusion rate remains small, and the SL appears to be decoupled from the rest of the atmosphere, allowing for the high concentration buildup, as observed. Concentrations above 150 m show relatively homogenous profiles up to the maximum altitude with typically cleaner air. Given the atmosphere's stability during the first ascent, only a little or no vertical dispersion is occurring at these altitudes. Between the first ascent and the following descent, the surface temperature increased by 4.5° C

in response to incoming solar radiation. The temperature of the entire column also increased, and the main surface-based temperature inversion dissipated (11a). A shallow inversion layer of 1° C/100m can still be observed in the second profile between 50 and 75 m (orange) and the third profile (purple) between 100 and 125 m. However, atmospheric stability generally decreases between the first and last profile, inducing convective turbulence and entrainment of the residual layer into the surface layer. This phenomenon can be observed in Fig. 10c and 12, where the high concentration at the surface in the first

profile, indicated by the yellow colors, gradually decreased for each profile. The surface dilution is observed for all tracers, and by 11:00, all profiles appear rather homogenously distributed up to the maximum reached altitude. The efficient mixing effectively reduces particle and gas concentrations near the surface and alleviates air quality issues. The observed homogenous profiles suggest that the induced convective mixing and slope winds can transport polluted air from the surface to higher elevations, as previously reported by Furger et al. (2000) during the VOLTALP campaign in the Mesolcina valley in southern

Switzerland. Similar conclusions were drawn by Ketterer et al. (2014) who reported an increase in local boundary layer height



and transport of aerosols from the valley bottom to the Jungfraujoch by slope winds. Aerosol particles can potentially be transported into the free atmosphere if the convective activity develops sufficiently, with subsequent further transport over longer distances in the FT. Contrary to Harnisch et al. (2009), who observed that slope winds could split at higher elevations in winter because of elevated shallow inversions and bring the transported pollution back to the center of the valley creating

secondary pollution layers, we did not observe such a phenomenon. Results suggest that snow free slopes and stronger solar radiation in autumn allow for effective upward transport of valley bottom pollution compared to winter.

**4.2 Case 2 – Particle size distribution dynamics during the transition from a stable to a mixed boundary layer**

Fourteen profiles (7 ascents and 7 descents) were performed on a cloud-free day on October 14, 2021, from 06:50 to 12:30.

The instrumental setup of the flight is presented in Table 4. Figure 13 shows measurements at the surface and the altitude profile timeseries of the helikite. The altitude profile (Fig. 13d) shows an alternation of fast ascending, descending, and stepwise profiles. Stepwise profiles are typically performed during descents to increase sampling time at specific altitudes to run multiple scans with the mSEMS to obtain several PSDs, which is important in low concentration environments, where single PSDs can be noisy. Based on the integrated particle number concentration ($N_{8-280}$) of the mSEMS (not shown here) and

$N_{186-3370}$ (Fig. 13d, colored altitude profile dots) we distinguished three layers. A surface layer up to 70 m, an intermediate entrainment layer (EL) between 70 and 150 m with a negative particle concentration gradient, and an elevated layer above 150 m that we consider the residual boundary layer from the previous day, therefore, denoted as RL. A subset of collected temperature profiles, evenly spaced out and covering the whole flight period, has been selected to show the evolution of the atmospheric structure (Fig. 14). The numbered profiles are also indicated in Fig. 13d for more clarity. Figure 14a shows a

temperature inversion caused by nighttime surface radiative cooling (Fig. 13a). The positive temperature gradient up to the maximum reached altitude is indicative of stable boundary layer conditions. The SL and EL show gradients of 3.5° C/100m and 0.6° C/100 m, respectively. The temperature profile remained relatively stable until 09:45; after that, the entire column was warmed under the influence of solar radiation (Fig. 13a). Note that because of lower sun elevation and high surrounding mountain peaks, sunrise occurred roughly one hour later on October 14 than October 1.

Winds remained very low at the surface throughout the flights, with a slight dominance of easterly direction until sunrise. Wind direction then changed due to warming of southerly exposed slopes (Fig. 13a). The vertical wind profile indicates increasing northeasterly winds with altitude during the first profiles. However, winds decreased after 10:45 and were almost inexistent during the last profiles, indicative of a transitioning regime between katabatic and valley winds. Figure 13c shows the evolution of the SL. Despite the presence of a temperature inversion that developed overnight, the concentration in the

surface layer shows an evident increase after 07:15 (Fig. 13c) in response to increased traffic emissions. We then observe a dilution and a larger vertical extent of the SL after 10:00. After 11:30, the surface layer is not visible anymore.

Based on Fig. 13c and d, three periods have been identified. The first period [P1] (07:30 – 09:59) represents the accumulation of pollutants in the SL. From 10:00 to 11:15 [P2], we observe a slightly greater vertical extent of the concentrated layer,



indicative of a boundary layer development and ongoing vertical mixing. Finally, after 11:15 [P3], the profile is more
homogenous with no clear surface layer, consistent with the particle concentration decrease observed on Fig. 13c. Note that
although the total particle concentration shows a decreasing trend shortly after 10:00, a peak of particles was measured around
10:40. This sudden burst was probably related to a very close source of anthropogenic emissions from a truck or gardening
activities on the nearby parking lot. These nearby emissions might have biased to surface concentrations of the ascending
profile at 10:47.

For each period, we investigated the PSD measured with the helikite to identify the main characteristics of each layer and see
how they evolved with the development of the ABL. Results for PSD between 8 and 500 nm are presented in Fig. 15. The
distribution was obtained by merging data from the mSEMS and the POPS. The two datasets present an overlap between 186
and 280 nm. Note that no conversion was made to transform the optical diameter from the POPS into the electrical mobility
diameter. Left panels (a, c and e) show the color-coded evolution of the PSD in each layer. The SL is represented on the lower
panels for easier interpretation. Right panels (b, d and f) show the equivalent normalized distribution to better evaluate the
relative contribution of different size modes to the PSD. Normalization was done by dividing dN/dlogDp values of each scan
by the maximum dN/dlogDp measured for the respective scan, yielding a maximum value of 1 for the main peak.

The SL (Fig. 15e and f) is characterized by the highest concentration during P1 (yellow) and P2 (light brown). Looking at the
normalized distribution, the SL seems dominated by a small Aitken mode around 15 nm. A second mode is also visible during
P1 between 30 and 40 nm (small shoulder in the distribution). This second mode is also present on the upper layers and
represents most likely aged particles emitted during the previous days. At P2, this larger Aitken mode is not visible anymore
because of the stronger dominance of freshly emitted particles at the surface. Note the main peak at P2 (Fig. 15f) has shifted
to the right compared to P1, indicative of potential growth of freshly emitted particles. Looking at the RL (Fig. 15a and b), the
PSD exhibits a bimodal distribution with a main larger Aitken mode at 40 nm and an accumulation mode at roughly 150 nm.
This distribution seems to represent the background boundary layer composition of particles emitted from previous days
(Aitken mode) and older particles that either remained suspended in the ABL for longer or were entrained from the free
troposphere. At P1, the PSD also shows contributions from smaller nucleation mode particles. It can be hypothesized that
emissions from cars and residential heating on the valley sides could directly contribute to this increase of smaller particles in
the RL. The size distribution is, therefore, the result of the mixing between the aged mode from the previous day and fresh
emissions from higher up in the valley. At P2, the contribution of the nucleation mode is lower but with large variability,
indicative of a transition to lower car traffic on the valley sides. A more systematic analysis under similar conditions would
need to be performed to see if this phenomenon regularly occurs and better understand the underlying processes.

The EL shows a similar feature to both the SL and RL. At P1, the PSD shows more similarity  with the RL but with a less
pronounced Aitken mode peak (Fig. 15c and d).  At P2, the influence from the surface becomes clearer as the overall
concentration of nucleation and Aitken mode particles increases similarly to the SL. This indicates the onset of boundary layer
growth and upward transport of surface emissions. At P3 (dark brown), the EL and SL show very similar characteristics with
the same concentration magnitudes for a nucleation mode peak, the larger Aitken mode (40 nm) and the accumulation mode



with overall lower total concentration indicative of a larger mixing volume due to boundary layer growth. The observed increase in the nucleation mode contribution could be explained by a combination of NPF without growth and direct emissions

of ultrafine particles by cars. However, due to a limited amount of measurements in the layer, the actual source of the nucleation mode contribution remains uncertain. The RL shows similar features and concentration magnitudes as the lower layers for the Aitken and accumulation mode, but not for the nucleation mode, potentially indicating that these particles were only emitted later and did not have time to be transported higher up yet and where thus not captured. . The bimodal distribution observed in the former RL at P3 seems to constitute the background size distribution of the mixed boundary layer (ML) in the valley.

Overall, in the presence of a stable boundary layer, surface pollution is tightly linked to traffic emissions and is constrained in a shallow layer about 70 meters thick. This can lead to a rapid accumulation of pollutants. Ultrafine particles around 15 nm dominate the number concentration, which can be up to 5 times higher than the concentration of a mixed-boundary layer if we refer to the previous case study (Sect. 4.1). Part of these particles remain in the boundary layer after the development of a ML and grow to a size of about 40 nm. These particles then constitute the boundary layer's particle background along with particles

in the accumulation mode. The development of the ML in response to surface heating is fast, and the concentrated surface layer is typically diluted within 1 to 2 hours.

### 4.3 Examples of offline chemical analysis of airborne samples

Two test flights of airborne sample collection were performed on September 28 and October 7, 2021. For both flights,

MoMuCAMS was equipped with the HFI for aerosol chemical analysis, 8-channel filter sampler (FILT) for SEM and TEM analysis, and the POPS. The flight pattern for both flights was similar. After reaching the desired sampling altitude, the HFI pump was turned on remotely while the balloon hovered at the same altitude. Simultaneously, the FILT sampled for roughly 1 hour per channel. The aim of the system is to sample air from layers that are decoupled from the surface to assess and compare the aerosol chemical composition to the mixed ABL or SL. Such a strategy can be applied to assess the chemical

composition in decoupled layers above the surface when the lower troposphere is stratified. However, given the vertical extent of the daytime mixed ABL during the field campaign and the tether length, sampling was performed in the mixed ABL and constituted mainly a proof-of-concept of the sampling system. In both cases, the measured vertical profiles during ascent and descent indicated a well-mixed atmosphere with similar $N_{186-3370}$ concentrations throughout the entire column. The temperature profiles indicated an adiabatic lapse rate. An estimation of the aerosol mass concentration during sampling time was calculated

from particle size distribution measurements from the POPS, assuming a mean particle density of 1.6 g cm$^{-3}$. Flight 1 and 2 had average concentrations of 3.58 [1.43] and 1.48 [1.37] μg m$^{-3}$, respectively. The values in brackets indicate the standard deviation. Due to increased wind conditions (from 1.5 [2] to 9 [5] m s$^{-1}$ for flight 1 [2]) between the beginning and end of sampling, the altitude of the balloon decreased slightly. Table 5 provides details of both flights. Additionally, samples were also collected at the surface before flight 1 and, before and after flight 2 to obtain a ground reference.



Collected aerosols have been analyzed for element concentrations (see Sect. 3.6), and results for Cu and Se are presented here as an example. High concentrations of Cu and Se in fine particles can have adverse health effects through direct inhalation or direct exposure via deposition (Daellenbach et al., 2020; De Santiago et al., 2014; Fang et al., 2017). Apart from being toxic at high concentrations, Se is an important micronutrient for humans (Winkel et al., 2015). It has been estimated that up to 1 billion people worldwide have inadequate Se intakes, largely due to low concentrations in staple food crops (Combs, 2001).

Because the atmosphere is an important reservoir of Se (estimated between ~13,000 and 19,000 tons of Se per year; (Wen and Carignan, 2007)) supplying terrestrial ecosystems and food chains, it is essential to understand atmospheric Se cycling to predict atmospheric Se supplies to surface environments. Besides implications for human health, an important aspect of the Cu atmospheric cycle is the input of atmospheric Cu to aquatic systems (oceans), which can influence primary productivity and phytoplankton community structure (Yang et al., 2019).

Figure 16 shows results of samples collected on the ground (a and c) and during flight (b and d). Ground sampling was performed with 6 stages and an after filter collecting all remaining particles below the lowest cutoff, while flights were performed with 3 stages only (0.44, 1 and 2.5 µm). Due to the low detection limit for Se, Se could be detected in almost all filters collected at the ground (between 12 to 18 h sampling time) and during flight (over 5 h). Due to higher Cu background in filters and thus a higher detection limit, Cu could mainly be detected in filters collected at the ground. Only one Cu

measurement in the 1 – 1.4 µm range was above detection limit for the aerosols collected during flight. The main limiting factor is the small aerosol mass concentrations obtained for the flight samples, which results from low pumped sample flow and sampling time. Great care must thus be taken in future studies in term of sampling strategy to ensure that the amount of collected material is sufficient for chemical analysis.

Figure 16a-b indicates that Se is mainly contained in submicrometer particles, with highest concentrations being measured in

the 0.44 – 1 µm range. This result is consistent with one previous study looking at Se size distribution in aerosols collected at three sites in the Baltic sea (Dudzińska–Huczuk & Bolałek, 2007). Interestingly, in contrast to Se, Cu concentrations increase with increasing particle size (Fig. 15c). Although it is out of the scope of this paper to investigate the factors controlling Cu and Se concentrations in the aerosol particles, the difference between Se and Cu size distribution could be explained by different emission sources. Sources of Cu are typically metal industries, fossil fuel combustion, and abrasion of car breaking

pads (Schauer et al., 2006; Yang et al., 2019). Vaporized copper sulfate used in the treatment of vineyards (Eckert and Jerochin, 1982) may also constitute a relevant source for the study area. For Se, 60 % of the atmospheric inputs have been estimated to be of natural origin and mainly from the biological production of volatile Se compounds, which are quickly oxidized and incorporated into the aerosol phase (Wen and Carignan, 2007). Despite the importance of the atmosphere in the biogeochemical cycling of trace elements such as Cu and Se that are toxic or essential for humans and (micro)organisms, atmospheric data are

still scarce, and often limited to point-source sites, low temporal resolution (daily to weekly sampling), and/or one size fraction (PM2.5 or PM10). Our results show the feasibility of investigating aerosol composition with the adapted HFI on board of MoMuCAMS deployed at the ground or during flights. The MoMuCAMS system thus has great potential to improve our



understanding of aerosol sources and transport, which is of importance for various fields of environmental sciences including climate and trace element biogeochemistry.

**4 Conclusions**

This manuscript presents a newly developed system for tethered balloon observations of aerosols and trace gases in the lower atmosphere. MoMuCAMS is a modular system, that allows different instrumental configurations to combine aerosol microphysical, optical and chemical properties observations with trace gas composition measurements. To the authors' knowledge, this is the first time a tethered balloon system has been set up to measure a full aerosol size distribution from 8 nm

to 3 μm. This information allows us to better study the origin of aerosol particles, their physical and chemical transformation and transport at altitudes relevant to the Earth's radiative budget. MoMuCAMS has been designed to be deployed with a helikite, because of the balloon's rugged characteristics. It is ideal for flying in extreme weather, including windy and cold conditions. Therefore, it can be used in Arctic or Antarctic regions, where many questions remain regarding aerosol-cloud interactions and aerosol radiative effects. The system has already proven to remain very stable at winds above 15 m s$^{-1}$ and has

flown at temperatures as low as -36° C.

Because MoMuCAMS uses several relatively new instruments, laboratory and field characterizations have been performed to demonstrate the high data quality and related uncertainties. The inlet system was also characterized for sampling efficiency and transmission losses to ensure a complete description of the system. Two portable aMCPCs showed deviation below 5% from a reference MCPC. We tested the sizing accuracy and transmission losses of the mSEMS using PSLs of different sizes.

The maximum deviations of measured mobility diameters were 8% and 3.1% for 51 and 70 nm PSL, respectively, and below 1% for 150 and 240 nm PSL. Based on the particle transmission tests, it is important to correct the mSEMS size distribution for losses of smaller particles. The manuscript provides a first empirical correction function. Two POPS were tested for sizing and counting efficiency. Sizing accuracy remained between 10 and 20% up to 800 nm particles for the two instruments. To mitigate sizing errors for larger particles we decided to use a 16-bin size resolution (with 6 bins for particles larger than 800

nm). We also showed that the three smallest bins of the instrument are affected by spurious noise and should be excluded from the analysis, resulting in an effective cutoff size at 186 nm. The counting efficiency for particles larger than 186 nm for both POPS is within 10% from a reference CPC. No specific characterization was performed for the STAP, as it has already been well characterized for airborne observations (Bates et al., 2013; Pilz et al., 2022). The Mira Pico for CO measurements was presented and characterized for its two modes of operation. Finally, procedures for samples collection used for electron

microscopy and chemical analysis using ICP-MS/MS were presented. All these results demonstrate the suitability of the instrumental set up for airborne in situ measurements.

The MoMuCAMS has been tested during two field campaigns in the Swiss Alps, in January and September 2021 as well as in February 2022, in Fairbanks, Alaska, to study the vertical dispersion of air pollution in a sub-Arctic urban area in winter (ALPACA field study) (Simpson et al., 2019), and in September 2022, in Pallas, Finland, to study cloud formation (PaCE2022



field study) (Doulgeris et al., 2022). Three case studies from the September field campaign in 2021 in Brigerbad, in the Rhône valley, Switzerland featuring different instrumental setups have been presented here in detail in Sect. 4. Case studies from October 1 and October 14, 2021, showed a surface-based inversion in the morning providing an opportunity to test the ability of MoMuCAMS to observe aerosol and trace gas dynamics in evolving boundary layer conditions. The vertical structure of the ABL in the morning featured in both cases a surface layer with a top between 50 and 70 m above ground level, an

entrainment layer characterized by a negative gradient of pollutant concentration up to 150 m and a residual layer above. The surface layer build-up typically occurred during morning rush hours and was dominated by traffic emissions with a main particle size distribution mode around 15 nm. Total particle number concentrations (>7 nm) were up to seven times higher in the surface layer compared to the residual layer. We also observed an increase in ultrafine particles in the residual layer before the inversion breakup suggesting that traffic on the valley slopes constitutes a significant emission source into the residual

layer in the early morning hours. Following sunrise, the surface layer typically dissipated within less than two hours leading to efficient mixing in the ABL and homogenous vertical distributions of particles and trace gases. Additionally, the first case study featured an elevated narrow pollution plume between 80 and 125 m above ground level on the first ascent.

A third case study illustrated the capability of the system to perform aerosol sampling at a chosen altitude over several hours. The ability of the system to sample for long periods has shown to be beneficial especially in conditions of low concentrations,

where extended sampling is required to collect enough mass for chemical analysis. Collected samples can be used to provide size segregated chemical composition using mass spectrometry and/or SEM/EDX or TEM/EDX. The analysis of chemical composition and aerosol morphology at higher altitudes will allow us to tackle questions related to aerosols' origins (e.g., anthropogenic versus natural), and their physical and chemical transformations in the atmosphere. A deeper understanding of the aerosols' composition, size and morphology will also allow a better constraining of their impact on climate and ecosystems.

The MoMuCAMS system characterization presented here provides a reference for future studies and assures the reliability of the measurements performed with MoMuCAMS. The case studies show the potential of our platform for vertical measurements of aerosol sources and processes in the lower part of the troposphere. The system can be continuously developed to integrate different instruments and to relate the in situ vertical observations with ground-based remote sensing (e.g., with an aerosol lidar) or drones carrying a subset of instruments for a more complete characterization of the ABL's horizontal and vertical

structure.

Overall, MoMuCAMS is an easily deployable tethered balloon system able to cope with high wind speeds and cold conditions and to fly inside clouds, providing reliable and high signal to noise data. The advantage of the MoMuCAMS-helikite system over other airborne platforms is the ability to observe processes in situ over several hours without needing to move position, thereby providing insights that were difficult to obtain beforehand.




**Code availability**

The scripts used for the analysis in this study can be provided by contacting Roman Pohorsky (roman.pohorsky@epfl.ch).

**Data availability**

Data are freely available by contacting Roman Pohorsky (roman.pohorsky@epfl.ch).

**Author contributions**

JS conceived the original MoMuCAMS idea and obtained the funding. RP, AB and JS developed the MoMuCAMS system,

performed the different laboratory and field measurements. JT and LW developed the analytical methodology for ICP-MS/MS chemical analyses. RP and AB performed data analyses. RP wrote the manuscript with contributions from AB, JT and JS. All the authors commented on the manuscript.

**Competing interests**

The authors declare that they have no conflict of interest.

**Acknowledgment**

This work received funding from the Swiss Polar Institute (Technogrant 2019) and the Swiss National Science Foundation (grant no. 200021_212101). JS holds the Ingvar Kamprad Chair for Extreme Environments Research sponsored by Ferring Pharmaceuticals. The authors would like to acknowledge the work of Stéphane Voeffrey, Robin Délèze and Dennis Ellersiek for their contribution to the MoMuCAMS construction and Emad Oveisi for his assistance with the analysis of collected

samples with the electron microscopes. We would like to thank Elyssa Beyrouti and Mike Chan from Eawag for their support with the sample preparation for the elemental analysis. We also would like to thank the Extreme Environments Research Laboratory team for logistical and field experiment support.




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



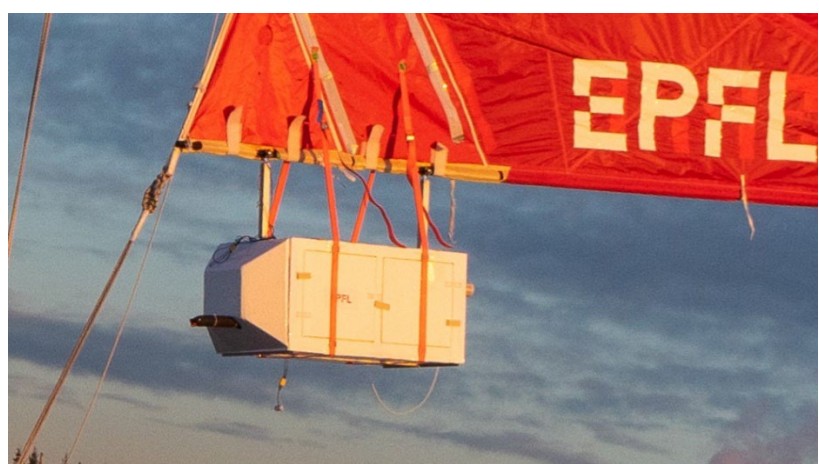

**Figure 1: Picture of the MoMuCAMS payload attached to the helikite. Two aluminum bars connected directly to the helikite's structure ensure stability of the payload. Two additional cargo straps provide additional safety for the payload attachment. The system remains very stable, even at winds above 15 m s⁻¹.**

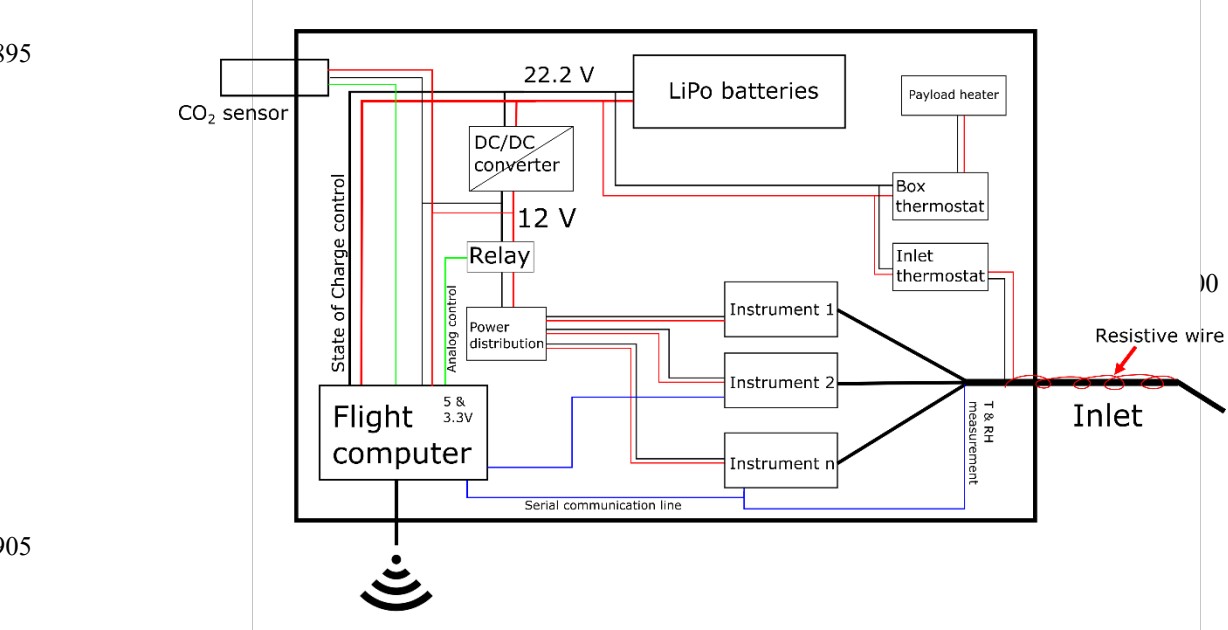

**Figure 2: Schematic of MoMuCAMS design. Black and red paths represent power wires. Blue and green lines represent serial and analog communication connections for communication between different instruments/components and the flight computer. The setup is flexible and can accommodate different aerosol and trace gas instruments, thus the layout of instruments is only illustrative.**



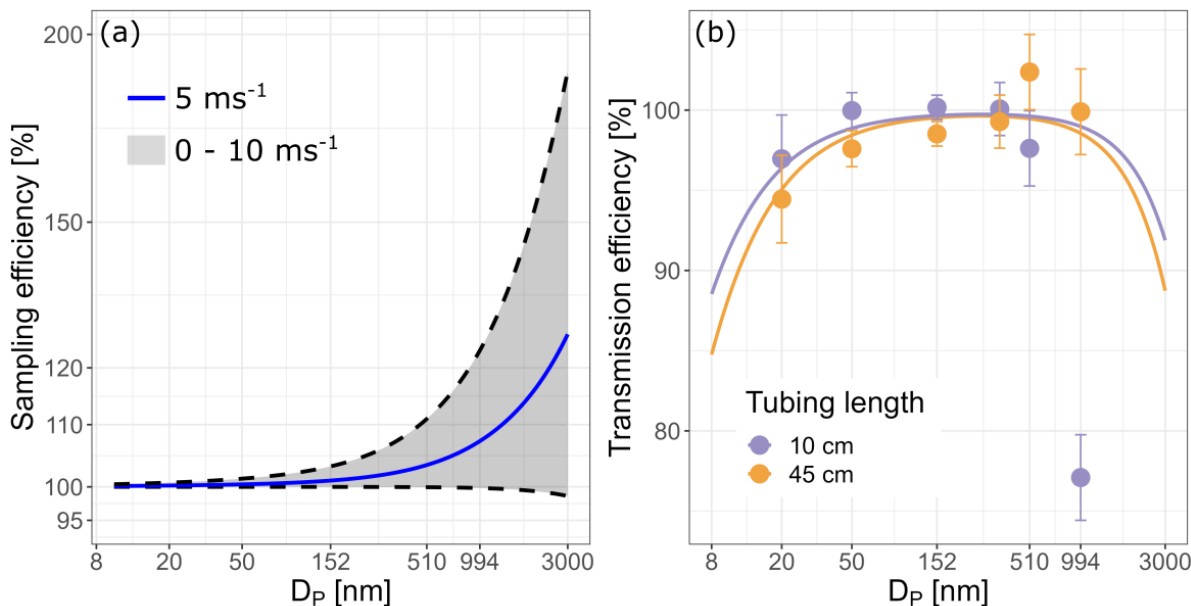

**Figure 3: a) Inlet sampling efficiency at 1.72 lpm sampling flow. The shaded area represents wind speeds between 0 and 10 m s⁻¹. The blue line represents the sampling efficiency at 5 m s⁻¹. b) Inlet transmission results from experimental tests and the PLC. Each dot represents a 5-minute average of transmission efficiency measurements and the error bars represent the standard deviation. The two lines are results obtained from the PLC. Colors indicate the length of the black tubing connecting the end of the stainless steel inlet to the CPC and represent the range of line lengths inside MoMuCAMS.**



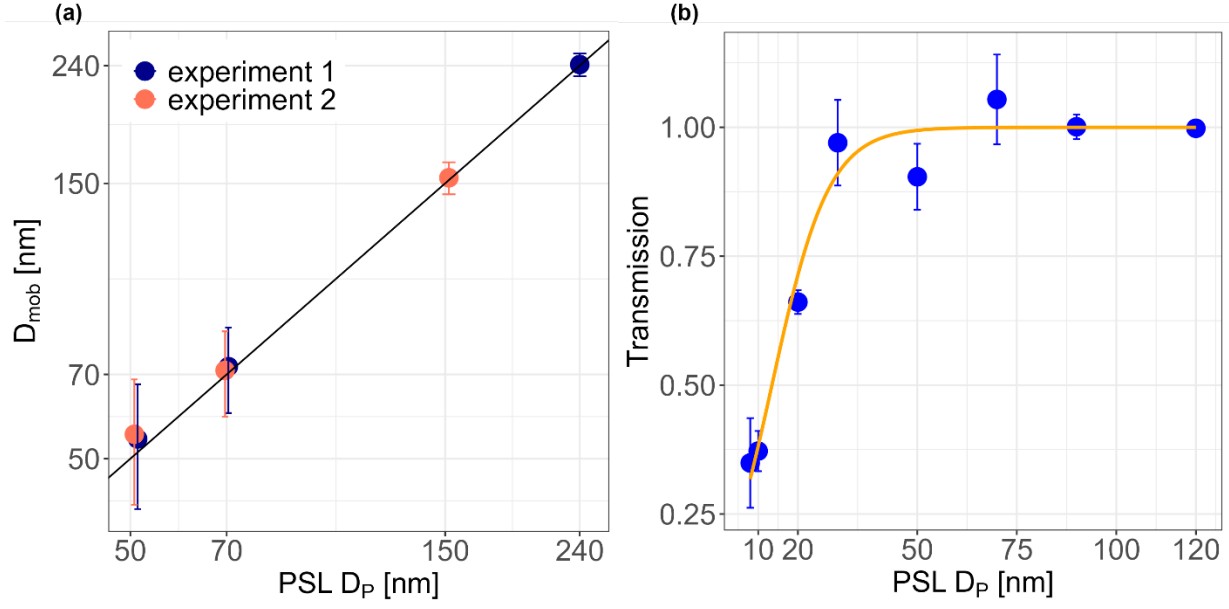

**Figure 4: a) Measured particle mobility diameter ($D_{mob}$) from a lognormal fit of the measured PSD from the mSEMS against PSL mean diameter. The black line represents equal diameters of PSL and measured $D_{mob}$. The experiment was conducted on two separate occasions (experiment 1 and 2). Uncertainty of the main mode is defined by one standard deviation of the lognormal distribution. b) Particle transmission through the DMA. Error bars indicate the standard deviation. The orange curve represents the best fit of the theoretical transmission function (Eq. 1).**

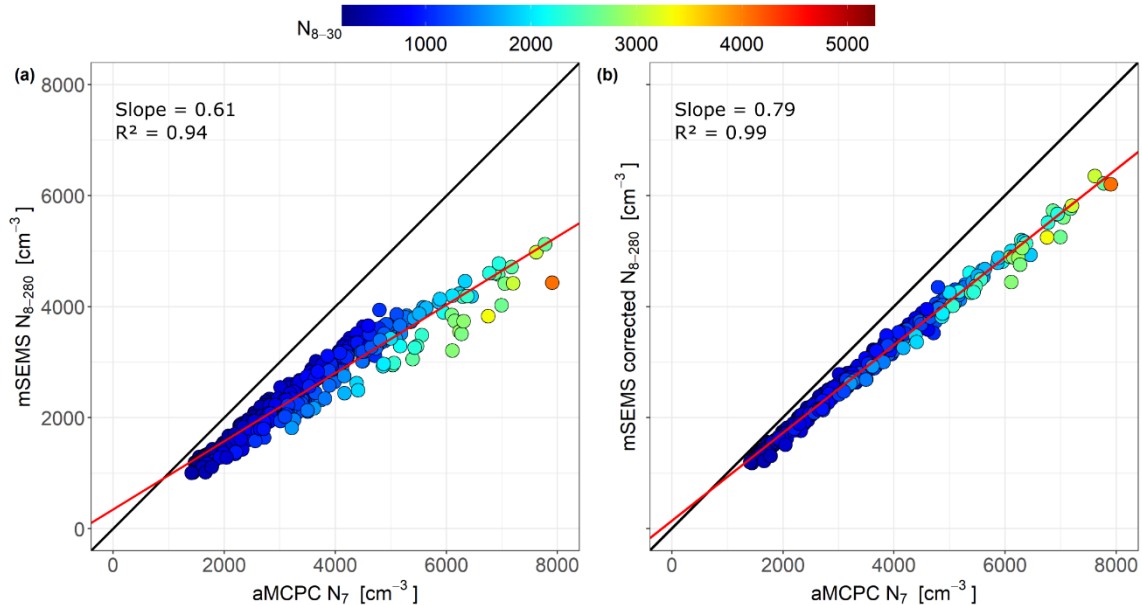

**Figure 5: Scatter plots of 10-min averaged particle number concentration. Panel (a) shows concentration from the aMCPC (x-axis) against the integrated measured concentration from the mSEMS (y-axis). Panel (b) shows the same but with corrected mSEMS data. The color scale indicates the total concentration of particles between 8 and 30 nm.**



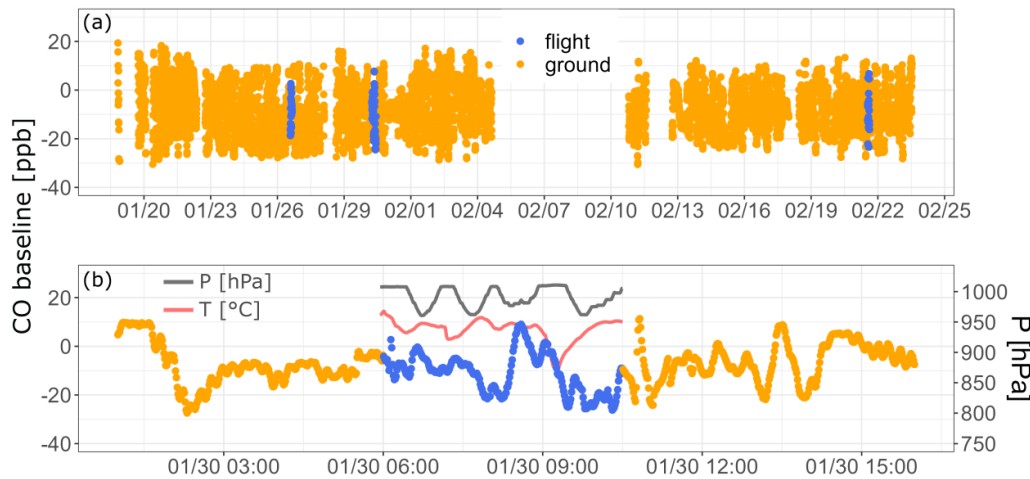

**Figure 6: a) CO baseline measurements of MIRA Pico during the ALPACA campaign from January 18 to February 24, 2022. Blue dots indicate measurements during flights. b) Subset of zero measurements before, during and after a flight on January 30, 2022. The black and red lines represent the barometric pressure (right axis) and temperature inside the box (left axis), respectively.**

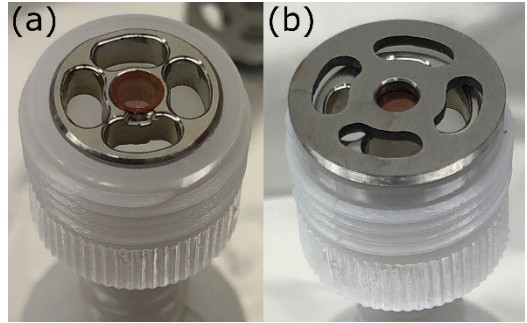

**Figure 7: a) TEM grid placed on custom-made grid holder. b) TEM grid with covering plate placed on top.**





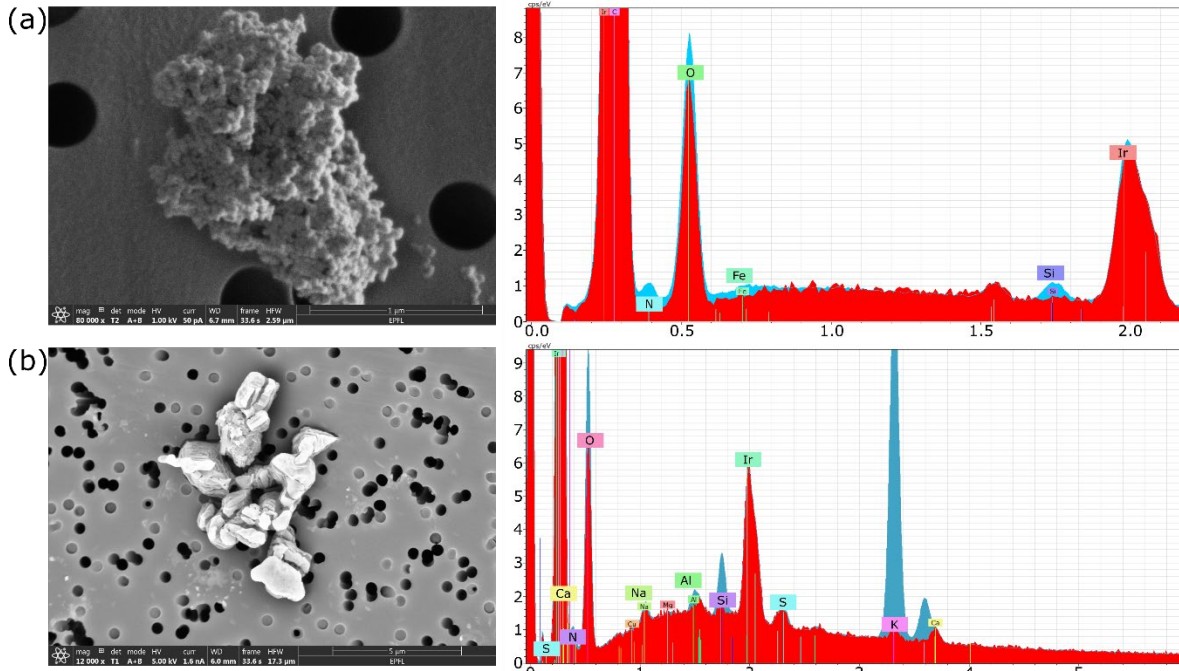

**Figure 8: SEM/EDX of two particles collected during airborne sampling on a) September 28 and b) October 7, 2021. Red spectra represent the EDX signal collected when pointing the electron beam only on the filter substrate which serves as a type of blank. Blue spectra indicate the EDX signal from the particle. (The SEM pictures were obtained in collaboration with Emad Oveisi, EPFL)**



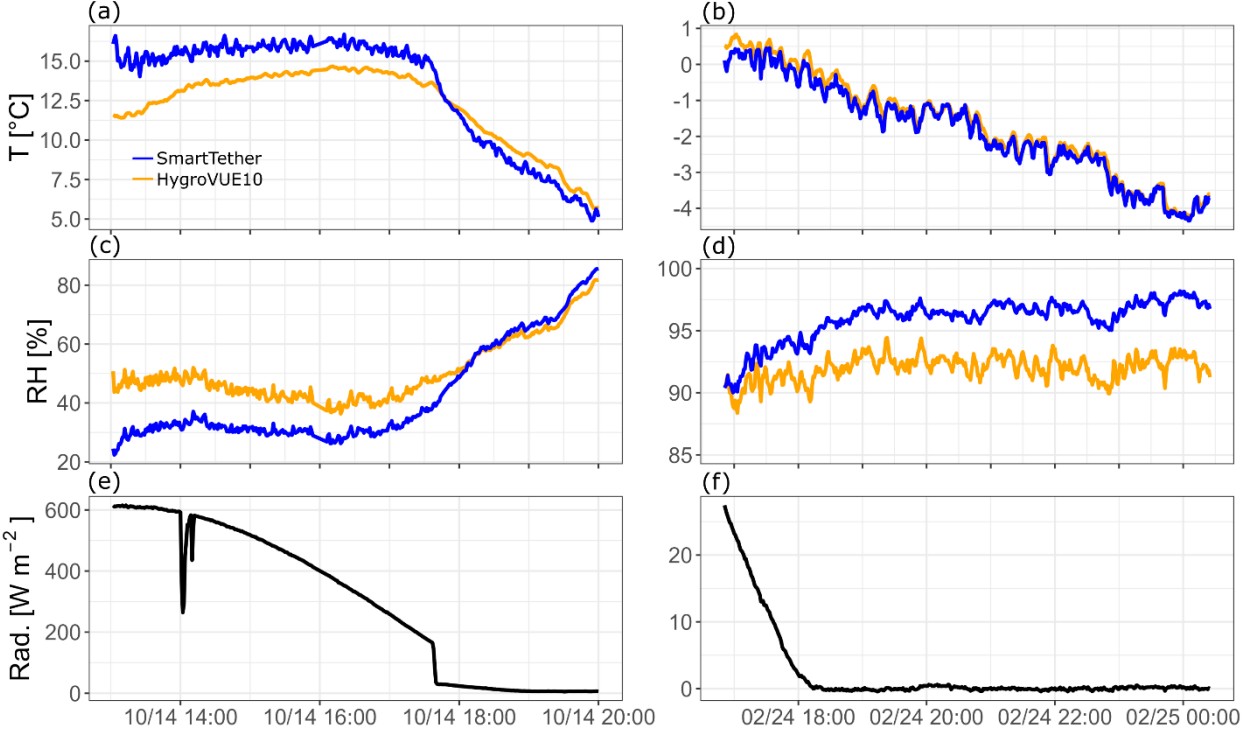

**Figure 9: Timeseries of temperature (T) (panels a and b), relative humidity (RH) (panels c and d) for the SmartTether (blue line) and HygroVUE10 reference sensor (orange line) during two comparison experiments (left and right columns). Bottom panels (e and f) indicated incoming shortwave radiation (Rad.) in black. Time is indicated in local time for both panels, CEST (left) and AKST (right).**

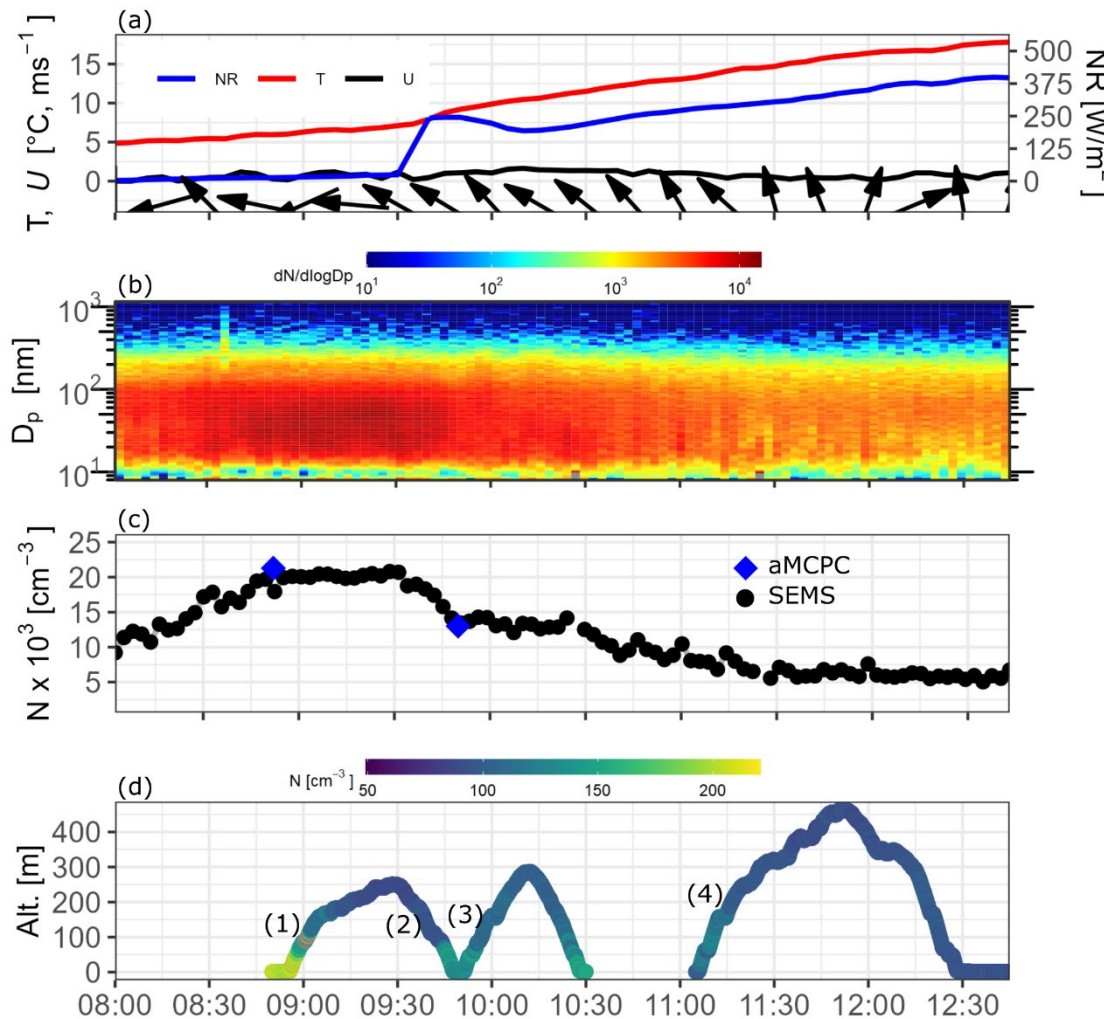

**Figure 10: Time-series on October 1, 2022 of (a) temperature (T), net radiation (NR) and wind speed (U) and direction (arrows) measured at the surface, (b) measured particle size distribution at the surface, (c) integrated total concentration (black dots) at the surface. Blue diamonds indicate the measured particle concentration ($N_7$) onboard MoMuCAMS when the helikite was at the surface, (d) balloon altitude above ground level [m]. The color scale indicates number particle concentration (> 186 nm). Numbers in brackets indicate the different profiles shown in Fig. 11 and 12.**



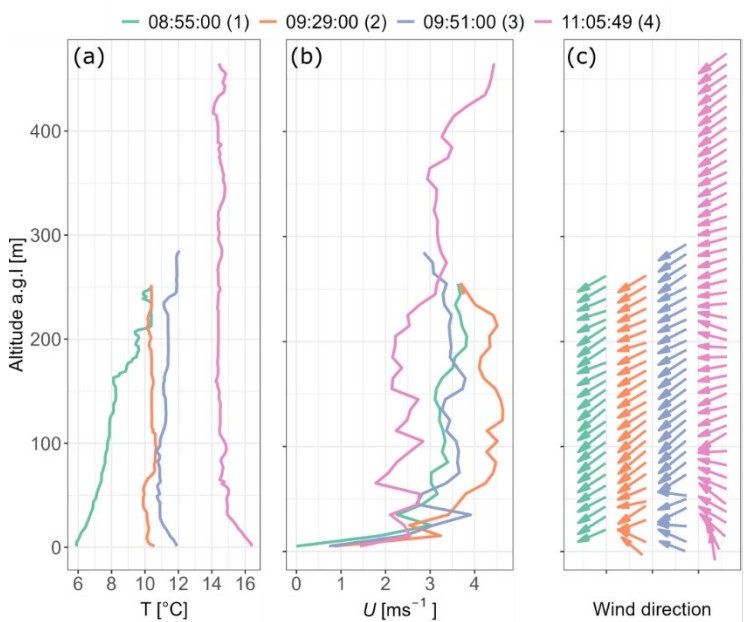

**Figure 11: Vertical profiles of (a) temperature (T), (b) wind speed (*U*) and (c) wind direction. Temperature is displayed at a 2-meter spatial resolution, corresponding on average to ten data points, whereas wind is displayed at a 10-meter resolution, for an average of 25 data points.**

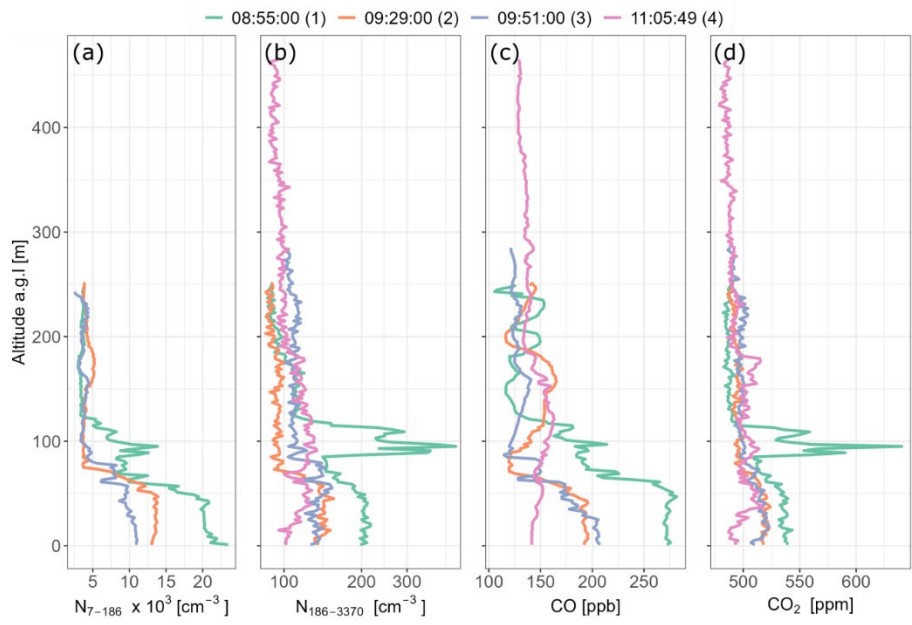

**Figure 12: Vertical profiles of (a) particle number concentrations in the size range of 7 to 186 nm, (b) particle number concentration in size range of 186 to 3370 nm, (c) CO mixing ratio, and (d) $CO_2$ mixing ratio. Data are displayed at a 2-meter spatial resolution, corresponding on average to ten data points. The displayed time on panel a) indicates the beginning of each profile.**





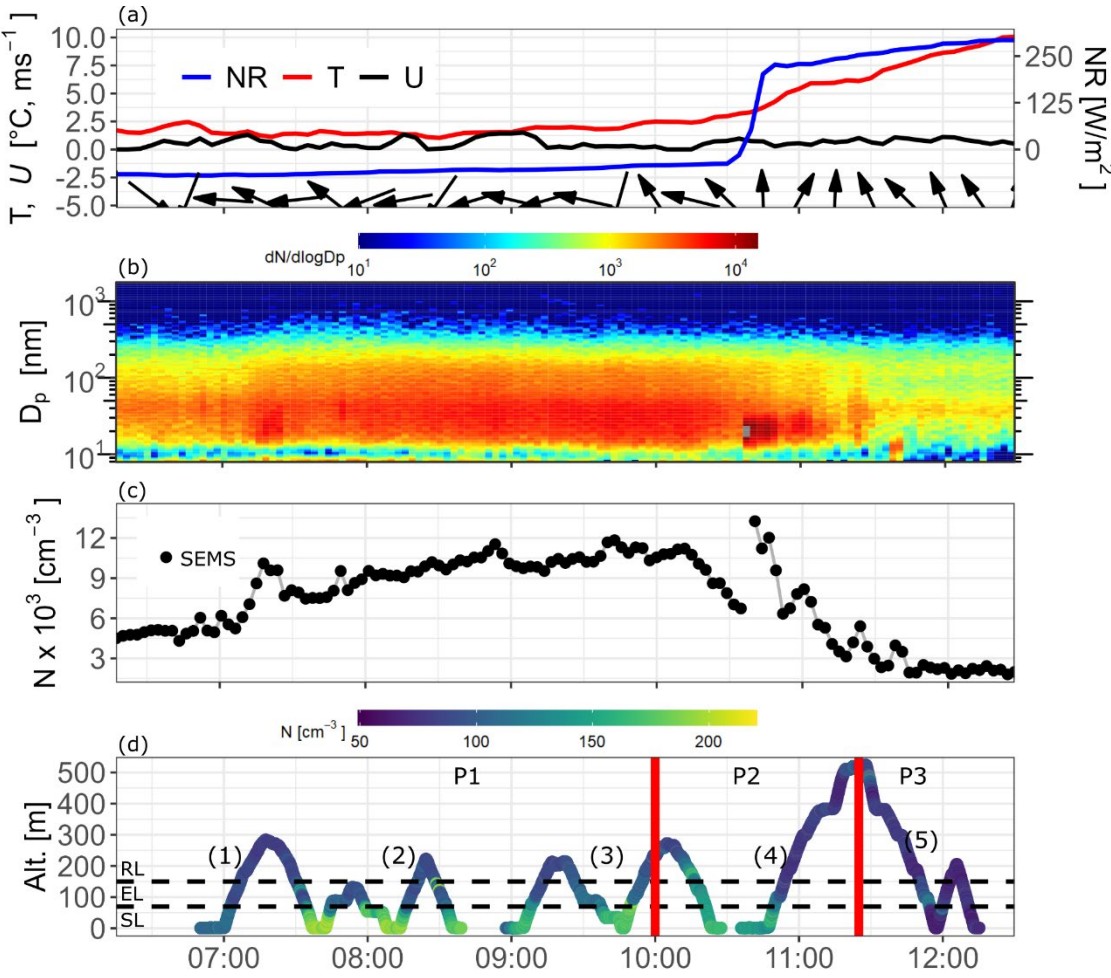

**Figure 13: Timeseries on October 14, 2022 of (a) temperature (T), net radiation (NR) and wind speed (*U*) and direction (arrows)**
**measured at the surface, (b) measured particle size distribution at the surface, (c) integrated total concentration at the surface and**
**(d) balloon altitude above ground level [m]. The color scale indicates particle number concentration (> 186nm). Numbers in brackets**
**indicate the different profiles shown in Fig. 14. P1, P2 and P3 refer to the three time periods discussed in Fig. 15.**



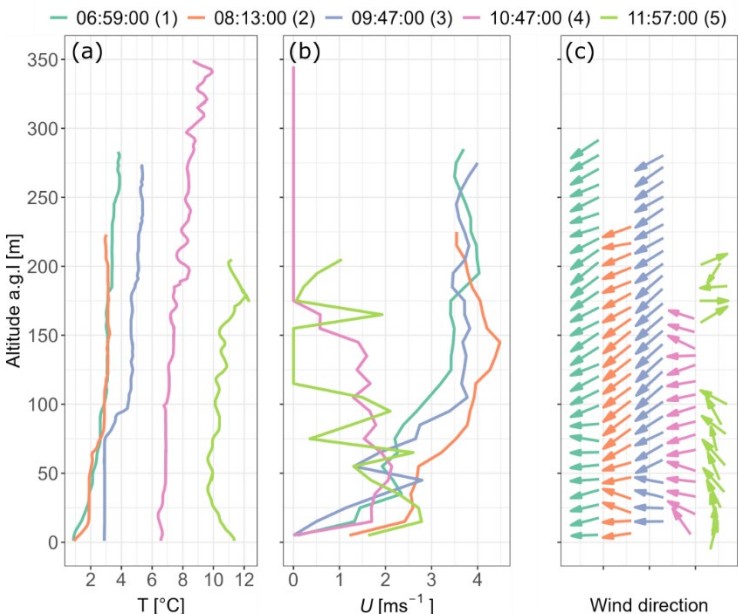

**Figure 14: Vertical profiles of (a) temperature (T), (b) wind speed (*U*) and (c) wind direction. Temperature is displayed at a 2-meter spatial resolution, corresponding on average to ten data points, whereas wind is displayed at a 10-meter resolution, for an average of 25 data points.**



**Figure 15: Evolution of particle size distributions between 8 and 500 nm in the residual layers (>150 m, a and b), intermediate layer (70 – 150m, b and e) and surface layer (0 – 70m, e and f). Solid lines indicate the median PSD measured by the mSEMS while shadings represent the interquartile range. Dashed lines represent the PSD measured by the POPS. Colors indicate the three periods P1, P2 and P3. Left panels (a, c and e) represent the dN/dlogDp size distribution. Numbers in the upper right corners indicate the number of scans collected per layer and period. Right panels (b, d and f) show normalized distributions where each dN/dlogDp value of a scan was divided by the maximum dN/dlogDp measured for the respective scan.**



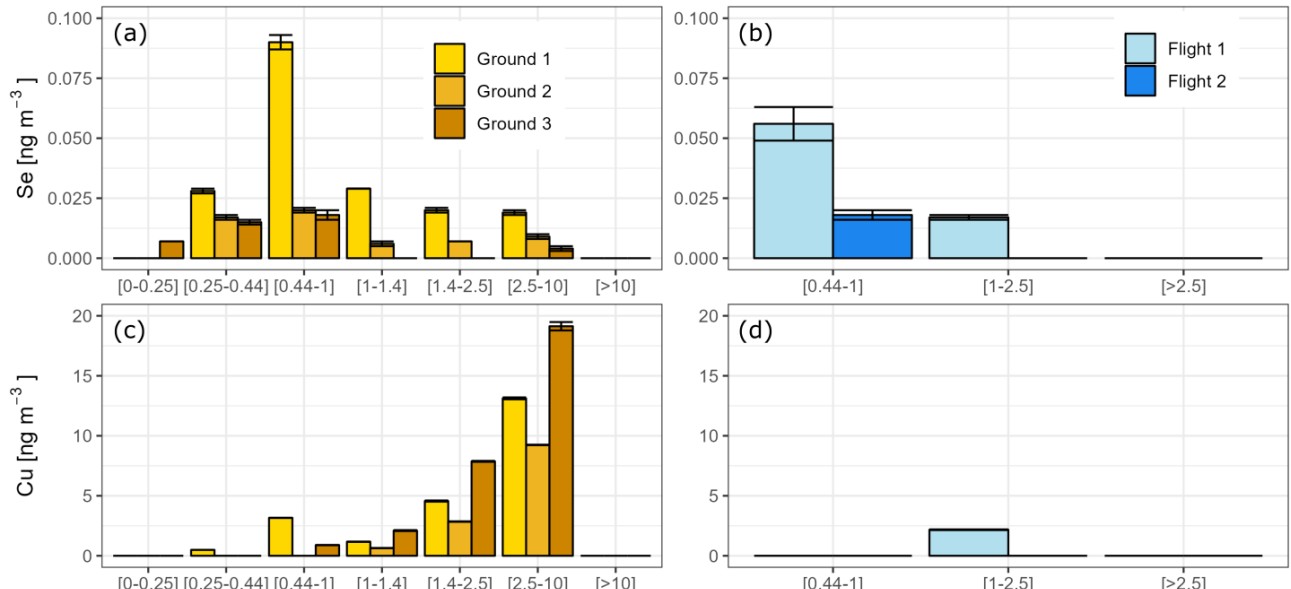

**Figure 16: Size segregated measured concentrations by ICP-MS/MS of selenium (Se) at the surface (a) and during flight (b) and of copper (Cu) at the surface (c) and during flight (d). The absence of a colored bar indicates that measured values were below the detection limit.**



**Table 1: List of instruments available on MoMuCAMS**

| Measurement / Analysis performed | Instrument | Manufacturer | Sampling flow (lpm) | Time resolution | Mode of operation |
|---|---|---|---|---|---|
| **Aerosols** | | | | | |
| Particle size distribution (186 – 3370 nm) | Portable Optical Particle Spectrometer (POPS) | Handix Scientific | 0.18 | 1s | 16 size bins |
| Particle size distribution (8 – 300 nm) | Miniaturized Scanning Electrical Mobility Spectrometer (mSEMS) | Brechtel Manufacturing Inc | 0.36 (0.1 – 0.76)* | 1s | 60 size bins / 1 sec per bin |
| Particle number concentration (7 – 2000 nm) | Advanced Mixing Condensation Particle Counter (aMCPC) | | 0.36 | 1s | - |
| Aerosol light absorption at 450, 525 and 624 nm | Single Channel Tricolor Absorption Photometer (STAP) | | 1.0 (0.5 – 1.7)* | 1 min | - |
| Microscopic analysis (SEM-EDX, TEM-EDX**) | 8-channel filter sampler (FILT) | | 1.5 (0.5 – 3.3)* | Adjustable, depends on mass concentrations, typically hours | e.g., 1 hour sampling per filter at constant altitude |
| Chemical analysis (IC, ICP-MS***) | HFI stage impactor Model 131A (MOUDI) | TSI | 100 | | |
| **Trace gases** | | | | | |
| $CO_2$ mixing ratio | $CO_2$ monitor GMP343 | Vaisala | (diffusion) | 2s | - |
| $O_3$ mixing ratio | $O_3$ monitor Model 205 | 2BTech | 1.8 | 2s | - |
| CO, $N_2O$ and $H_2O$ mixing ratio | MIRA Pico | Aeris Technologies | | 1s / 1 min | manual mode / differential mode |
| **Meteorology** | | | | | |
| T, RH, P, Wind speed and direction, lat, lon | SmartTether | Anasphere | - | 2s | - |

*Values in brackets represent the range of possible sampling flows, while the single value indicates the typical flow set during operations.*



*\*\*SEM-EDX = Scanning electron microscopy with energy dispersive x-ray analysis, TEM-EDX = Transmission electron microscopy with energy dispersive x-ray analysis (the analysis is done in laboratory after the flights).*

*\*\*\*IC = Ion chromatography, ICP-MS = Inductively coupled plasma tandem mass spectrometry (the analysis is done in laboratory after the flights).*

**Table 2: Summary table of background concentration values (in ng) for copper (Cu) and selenium (Se) extracted from 7 blanks (4 regular blanks and 3 field blanks) digested and measured by inductively coupled plasma tandem mass spectrometry analysis (ICP-**
1000 **MS/MS). The last column indicates the obtained detection limit calculated as the mean plus three standard deviations.**

| Element | Mean (ng) | Standard deviation (ng) | Detection limit (ng) |
|---------|-----------|-------------------------|----------------------|
| **Cu** | 8 | 5 | 22 |
| **Se** | 0.05 | 0.02 | 0.12 |

**Table 3: Meteorological parameters measured with SmartTether.**

| Measurement | Sensor (model, manufacturer) | Unit | Resolution | Accuracy | Range |
|-------------|------------------------------|------|------------|----------|-------|
| Pressure (P) | MS5540C, Intersema | hPa | 0.1 | 0.5 | 0 - 1100 |
| Temperature (T) | DS18B20, Maxim Integrated | ° C | 0.125 | 0.5 | -55 - +125 |
| Relative humidity (RH) | HIH9131, Honeywell | % | 0.1 | 1.7 | 0 - 100 |
| Wind speed (WS) | - | m s$^{-1}$ | 0.1 | 0.1 | 0 – 59 |
| Wind direction (WD) | - | ° | 1 | 2 | 0 - 359 |



**Table 4: Measured variables during flights**

|  | 01/10 | 14/10 |
|---|---|---|
| **Particle Number Concentration (>7nm)** | x* | |
| **Particle size distribution (8-270 nm)** | | x |
| **Particle size distribution (186-3300 nm)** | x | x |
| **CO** | x | |
| **CO$_2$** | x | x |
| **O$_3$** | x | |
| **Meteorological parameters (T, RH, P, WS, WD)** | x | x |

*aMCPC was removed for the 3$^{rd}$ profile.

**Table 5: Summary of ground and flight filter sampling.**

|  | Date | Mean sampling altitude above ground [m] | Altitude standard deviation [m] | Sampling time [h] | MOUDI sampled volume [m$^3$] | Number of collected filters for SEM | Number of collected filters for TEM |
|---|---|---|---|---|---|---|---|
| **Flight 1** | 09/28 | 279 | 59 | 5 | 30.2 | 3 | 2 |
| **Flight 2** | 10/07 | 434 | 47 | 4.85 | 28.9 | 3 | 3 |
| **Ground 1** | 09/27 | 0.6 | - | 17.9 | 107.4 | - | - |
| **Ground 2** | 10/06 | 0.6 | - | 17 | 102.1 | - | - |
| **Ground 3** | 10/07 | 0.6 | - | 12.7 | 76.1 | - | - |