# Peer review of "MoMuCAMS: A new modular platform for boundary layer aerosol and trace gas vertical measurements in extreme environments"

_EGUsphere, 2023_

## Author Comment (AC3)

We thank the editor and reviewers for the handling and commenting of our manuscript. Please find below our point-by-point responses to each of the reviewers' comments, including the modifications we have made to the manuscript. Reviewer comments (RC) are in black text and the answers to reviewers (AR) are given in blue text, and excerpts of the revised manuscript are given in *blue italic text.*

**Answers to Reviewer 1**

Anonymous referee #1, 11 May 2023

General comment

Pohorsky et al. present in their manuscript a newly developed tethered balloon platform for in situ measurement of atmospheric aerosol particles and trace gases. With a helikite and a modular suite of instruments, the so-called MoMuCAMS system can provide observational capabilities for particle microphysical and chemical properties in the lower troposphere. Development efforts were taken to allow deployments in challenging environments like the polar regions. Airborne measurements of aerosol vertical distributions are highly relevant for various studies and model evaluations, particularly in remote regions. Therefore, the work can potentially add a valuable contribution to the atmospheric measurement community and is suitable for publication in AMT after major revisions.

Parts of the manuscript need to be rewritten and restructured to improve readability. More accurate descriptions and carefully selecting synonyms should be considered to achieve correctness. The introduction, methods, and results sections can profit from a clearer separation; for instance, no results in the method section or background in the results section. The possibility of providing information not directly related to the platform within the supplementary should be considered. Conversely, some methodical details should be provided with the main manuscript instead of the supplementary. Comparisons of the mobile instruments of the platform with stationary instruments over a longer time range should be added for all instruments to prove the platform's capacity to provide quality-assured data. The authors are encouraged to resubmit after a comprehensive revision.

Thank you for providing these comments, it has helped us to greatly improve the manuscript.

Detailed comments

Abstract

RC1: Length should be reduced to the important content of the paper, redundancy should be avoided, and aspects not shown in the paper should not be claimed.

Since this is a technical paper, the focus should be on instrument performance and not too much on the case studies.

AR1: The abstract has been shortened to keep the focus on the technical aspect of the paper and avoid redundancy. In particular, the description of the case studies outcomes has been shortened.

RC2: Line 17: "multiplatform compatible" is redundant since it is part of the platforms name and abbreviation

AR2: The sentence has been reworked to get rid of the redundancy and reads as follows:

*MoMuCAMS has been primarily designed to be attached to a helikite, a rugged tethered balloon type that is suitable for operations in cold and windy conditions.*

RC3: Line 19: add "atmospheric" before boundary layer

AR3: Done.

RC4: Line 19 to 21: "These regions are known to …" no background is needed here and the properties are not specific for the mentioned regions

AR4: The sentence was removed.

RC5: Line 24 to 27: "This flexibility …" this sentence seems inaccurate because the multiple aerosol properties can only be measured in different configurations and not simultaneously. "cluster ions" and "optical properties" measurements were not shown in the paper. Chemical analysis is done offline, it's basically a balloon-borne filter sampler provided with the subsequent analysis being free to the user/application.

AR5: The flexibility doesn't imply that all measurements can be performed at the same time. However, it implies that the instrumental setup is easily modifiable and can be adapted to research. We consider this to be an important feature that allows us to use the same system to address different research questions requiring different suits of instruments. Moreover, although not all instruments can be flown at the same time, we still have the capacity for simultaneous measurements as the system has the capacity to carry several instruments at the same time.

Since our cluster ion counter has not yet been fully tested, we have removed it from the text. Light absorption is measured with the STAP and even if the paper does not show any results, it is important to mention it so the reader knows it is a feature of the system.

RC6: Line 27: "To the author's knowledge,…" should be avoided, leave it to the reader to decide

AR6: It has been removed.

RC7: Line 28: "full size distribution" is inaccurate, smaller and larger particles exist beyond the observational range, you can use "wide range" instead. The upper end of the size range should be mentioned and precisely define if you mean number or volume or surface size distribution

AR7: The sentence has been reworked as follows:

*It is the first tethered balloon-based system equipped with instrumentation providing a size distribution for aerosol particles within a large range, i.e. from 8 to 3370 nm, which is vital to understanding atmospheric processes of aerosols and their climate impacts through interaction with radiation and clouds.*

RC8: Line 32: the Finland campaign is actually not shown in the manuscript

AR8: Although direct results of the campaign are not shown but we consider it important to inform the reader of the deployments of the system for future scientific outcomes. We however now also included figures in the SI (Fig. S2 and S3) to illustrate the performance of the system under high wind and wet conditions that were obtained during the Pallas campaign.

RC9: Line 34: -36 °C and winds up to 15 m/s are not shown in the manuscript and shouldn't be claimed without prove

AR9: We have now included a figure showing a vertical profile with winds up to 15 m/s (Fig. S3) and an analysis of the horizontal displacement of the helikite as a function of windspeed (Fig. S3). As

these figures are only here to support our claims and do not provide any additional scientific value to the paper, we decided to keep this figure in the SI.

RC10: Line 35: "full characterization" seems inaccurate; full might mean something else for the authors than for others, and some important parts of the characterization are actually missing (see below)

AR10: We agree that the word full is inaccurate and it has been therefore removed.

RC11: Line 42: seems inaccurate, after the development of a mixed ABL there is no surface layer anymore

AR11: Thank you for noticing this inconsistency. We intended to say that the surface layer was rapidly eroded and followed by the development of a mixed boundary layer.

The new sentence reads as follows:

*Following sunrise, turbulent mixing led to erosion of the SL and development of a mixed boundary layer within one to two hours.*

RC12: Line 50 to 52: seems inaccurate, the presented system is not capable of capturing aerosol-cloud interactions. In general, a tethered balloon can only provide a snapshot from one position of a usually moving air mass and has, therefore, only limited process observation capabilities. The measurements always need to be put in context with other observations or modeling to use for process studies.

AR12: We take good note of the reviewer's comment and agree that our system cannot perform Lagragian observations. However, in the closing remarks of the abstract, we are not claiming that the MoMuCAMS can be used to investigate any type of aerosol process but simply that it can provide new insights compared to other systems. In fact, despite our system being localized, it enhances station-based observations by adding a vertical dimension and offers a time advantage over systems like drones due to its capability for prolonged operations. These two attributes can provide new insights into aerosol processes in the lower troposphere. Specifically, it supports larger payloads and extended flight times. This capability allows for comparative aerosol studies across different altitudes. For example, it can help in understanding if clouds form from local or transported aerosols. Additionally, our system can be used to investigate the occurrence of new particle formation in elevated layers and the potential entrainment in the boundary layer. While this process has been hypothesized to be significant across various global regions, direct measurements to confirm it remain sparse (Zheng et al., 2021; McCoy et al., 2021).
We also agree that measurements definitely need to be combined with remote sensing or modeling data. Typically we deploy the helikite together with ground-based lidars or ceilometers.

Introduction

RC13: Readability can be improved by restructuring the text to guide the reader only in one direction, from the broad topic of aerosol measurements to the specifics of balloon-borne observations with a technical focus. Jumping forth and back from detail to broad should be avoided

AR13: This is a very helpful remark. The introduction has been reworked and shortened to keep the focus on the technical aspect. We put this work in perspective and summarize the different measurement techniques available to make our point about the usefulness of tethered balloon systems.

RC14: Line 66: What do you mean by "strongly stratified"? Do you mean multiple atmospheric layers or stable stratification? The ABL often features multiple layers and stable stratification, not only in polar or mountain regions.

AR14: The term stratified was intended to describe a stable boundary layer. The paragraph has been modified to be more accurate. Although the stable boundary layer is not specific to polar and mountain regions, they are commonly observed there and represent places of interest from a climate and air quality perspective.

The paragraph has been reworked to read as follows:

*Understanding the vertical distribution becomes particularly important in environments where the atmospheric boundary layer (ABL) is highly stable. Poles and alpine valleys are two regions were a stable boundary layer is commonly observed (Chazette et al., 2005; Graversen et al., 2008; Harnisch et al., 2009; Persson et al., 2002). The stability leads to the layering of aerosols and reduced exchange processes, meaning that ground-based measurements are often not representative of cloud-level aerosol (Brock et al., 2011; Creamean et al., 2021; Jacob et al., 2010; McNaughton et al., 2011).*

RC15: Line 67: More precision required, stratification is a synonym for layering, do you mean "stable atmospheric stratification"?

AR15: See previous answer (AR14).

RC16: Line 75: "However, for assessing the direct and indirect radiative impact of aerosols, knowing their vertical distribution is vital." was written almost the same in line 63

AR16: Thank you for pointing out this repetition. Since it has already been explained above, this sentence has been removed.

RC17: Line 80 to 83: seems inaccurate, some of the described circumstances only belong to polar regions.

AR17: This was indeed inaccurate and the paragraph was just supposed to refer to polar regions. The word "mountain" has been removed. The text reads now as follows:

*Shortcomings are particularly large in polar regions, where space-born aerosol-focused remote sensing (e.g., Cloud-Aerosol Lidar and Infrared Pathfinder Satellite Observation, CALIPSO) provides nearly no data north of 82°N, signals become attenuated under thick clouds, sensors are challenged by surface brightness, and aerosol concentrations are often too low (Kim et al., 2017; Mei et al., 2013; Thorsen & Fu, 2015).*

RC18: Line 84: "shallow inversions" seems inaccurate, an elevated inversion layer that is in the range of the lidar can also be very shallow. Do you mean "shallow ABL"?

AR18: We modified the text to specify that we refer to surface based inversion, to avoid any confusion.

*Ground-based remote sensing is limited in vertical resolution, because retrievals do not start at the surface but further aloft, which is a key problem in regions with very shallow surface based temperature inversions.*

RC19: Line 88: "air layers" is very inaccurate, do you mean "atmospheric layers" ?

AR19: We changed "air layer" to "atmospheric layer".

RC20: Line 88 to 89: "Moreover, typically aircraft do not fly within the first hundreds of meters above the ground, missing therefore valuable information". This statement seems inaccurate. Many different kinds of aircraft can perform manifold flight patterns depending on the research objective, as shown in various studies. The advantage of balloons is that they can provide higher resolution, operate under icing conditions, and be within supercooled clouds for an extended time.

AR20: It is true that different types of aircraft can perform different types of flights, although for most of them, if not all, low level flights in low visibility is not possible. The sentence has therefore been modified and reads as follows:

*Moreover, an aircraft is typically limited for low altitude flights, especially under low visibility and icing conditions.*

RC21: Line 95: seems inaccurate, HOVERCAT is not a tethered balloon system

AR21: That is correct. We have therefore removed the HOVERCAT from the list of examples.

RC22: Line 107: "To the authors' best knowledge…" see above

AR22: Removed

RC23: Line 109: since the platform is designed for arctic regions, it should be stated that CCN can be well below 100 nm in size

AR23: We added the following sentence:

*It should also be noted that in the specific context of polar regions, CCN can be well below 100 nm in size (Schmale et al., 2018).*

RC24: Line 116: "…to be deployed on sea ice" is true, but it is not a proper criterion. Way heavier instruments and infrastructure have been deployed on sea ice. Usually, sufficiently thick sea ice floes are selected for scientific operations

AR24: That is true. The point that we wanted to make here is that the full system requires reduced logistics compared to larger tethered-balloons or aircraft and can therefore be deployed without too much assistance. However, this part has been removed from the introduction as it is not completely specific to MoMuCAMS and to reduce the overall length of the introduction.

Section 2

RC25: Provide the model, manufacturer, and country in brackets behind the devices that occur in the text e.g. helikite (Desert Star, Allsopp…, GB)

AR25: Done

Section 2.1

RC26: Please provide a more detailed mechanical sketch of the payload enclosure to prove the claimed "flexibility" of the system to accommodate multiple instruments. Figure 2 doesn't provide enough details on that

AR26: Done. The figure has been adapted. We included a 3D drawing of the box (Fig. 2) sjowing the fixed elements of the system and the available space (with dimensions) to place instruments.

RC27: Provide measurements of the inner temperature of the box during low-temperature conditions

AR27: Fig. S2a in the SI shows a vertical profile of this flight with the ambient temperature, sampled air temperature and MoMuCAMS inner temperature.

RC28: Line 126 to 127: the paper actually shows the deployment of 8 different instruments in 3 different combinations, I wouldn't consider this "a very large number of combinations"

AR28: Technically speaking, given the amount of instruments listed in Table 1, and future acquisitions, it is indeed possible to have many different setup combinations. However, we agree that the term is perhaps not appropriate and has been changed to "various". The new sentence reads as follows:

*The novelty of this platform lies in its flexibility to accommodate various combinations of instruments within the weight and dimension limits.*

Table 1:

RC29: - measurements not shown in the paper e.g., the STAP, shouldn't be part

AR29: We take note of the reviewer's comment. However, we believe that as for description of the system, it is important to inform the reader of the different instruments that are flown regularly on MoMuCAMS. Since the instrument has already been characterized in existing literature, we decided to not show any characterization results here to reduce the overall length of the paper and not reiterate previously accepted work.

RC30: - add uncertainties

AR30: Done

RC31: - add weights of individual instruments since it is the limiting factor for possible combinations

AR31: Done

RC32: - a scientific objective of typical instrument combinations would be interesting

AR32: Since we present different instrumental combinations in Sect. 4, we introduced them at the beginning of each case study as suggest in RC77. Each instrumental setup is followed by a short remark on a scientific objective associated to the latter.

RC33: Line 129: "guest instrument" sounds awkward, it is sufficient to phrase it with additional instruments

AR33: The sentence has been shortened and reads as follows:

*Importantly, MoMuCAMS can easily be adapted for additional instruments.*

RC34: Line 141: use "transmission" instead of "streamed"

AR34: Done

RC35: Line 143: "subset of data", please specify

AR: The data subset is the same as listed above (line 145). The sentence has been modified for more clarity. The word subset has been removed. It now reads as follows:

*The data is visualized live on a graphical interface, which helps for decision-making and sampling strategy adaptation during flights.*

RC36: Line 145: "various instruments", please specify

AR36: To address this comment, this part has been rewritten to:

*Additionally, the operator can use the graphical interface to send commands to the MoMuCAMS microcontroller. Commands include activation and filter position change of the FILT, activation and flow control of the high flow impactor, activation of a relay to power a specific instrument at a desired altitude and general shutdown of the system.*

Section 2.2

RC37: Line 150: what is the maximum length of the tether, including the extension? This is an important limitation of the observational capabilities

AR37: We believe that there is a confusion about the word "extension", which in reality refers to the total length of the line. The dyneema will however not stretch significantly, even at the highest tensions produced by wind and any extension is therefore negligible. The maximum reachable altitude is therefore 800 m a.g.l if there is now wind. To avoid any confusion, we replaced the word "extension" by "length".

*The total line length is 800 m.*

Note also that MoMuCAMS is independent from the helikite and the system could be used with a larger helikite and longer tether, or any other type of tethered balloon. A remark has been added at the end of Sect. 2.2:

*While in this manuscript we focus on the system built for a 45 m³ helikite going to a maximum height of 800 m, MoMuCAMS is independent from the lifting platform and can be used with a larger balloon and longer tether to reach higher altitudes.*

RC38: Line 154: Please provide a time series of the flight(s) on which the extreme temperatures and wind speeds were observed

AR38: Please refer to our previous answer to RC9. (Figure S2)

Section 3

RC39: Please introduce abbreviations in the text at the first occurrence

AR39: Done

RC40: Line 165: Only instruments shown in detail in the manuscript should be mentioned, thus delete the STAP

AR40: Please, refer to our previous answer to comment RC29.

RC41: Line 172: Is the 30° bend sufficient for preventing water droplets from entering the inlet? What about the tilt of the tether and the balloon itself at higher wind speeds, can that be up to 30° so that the inlet is actually horizontal? Please discuss

AR41: Yes, the 30° has shown to be enough to prevent any water from entering the inlet. The tilt of the balloon is independent of the tilt of the tether. At higher wind speeds, the wind tends to push the sail, which is located at the back, upward, actually maintaining the payload in a horizontal position. Careful inspection of the inlet after each flight has never shown any signs of water infiltration in the sampling line.

The following sentence has therefor been added in the text:

*Careful inspection of the inlet after each flight has not shown any signs of water infiltration in the sampling line.*

RC42: Line 173: Provide a time series that shows the relative humidity behind the inlet is sufficiently reduced by the heating system on a flight in high-humidity environments

AR42: Fig. S2 in the supplementary material shows the ambient and sampled air relative humidity for a flight with high ambient RH.

RC43: Line 197 to 201: The explanation given for the large discrepancy between the 10 cm and 45 cm tubes for 510 and 994 is not sufficient. At the reported average number concentrations of 20 and 35 cm-3, the CPC should not show a deviation up to 25 % at 5 min averaged data. It appears that the measurements were not done properly or the CPCs had a malfunction.

AR43: After careful investigation, we could not identify a specific cause for the observed discrepancies besides the mentioned issues related to the low and unstable concentration. We therefore reproduced the experiment. In order to be able to produce more particles in the accumulation mode and especially in the coarse mode, we used Di-Ethyl-Hexyl-Sebacat (DEHS) and size selected the particles with an aerodynamic aerosol classifier. Results show much better agreement with the theoretical prediction. The new measurements for particles larger than 500 nm have been included in the Fig. 3 and the text has been adapted as follows.

*Transmission losses in the inlet have been experimentally tested with particles of different diameters ($D_P$). For particles up to 350 nm, polystyrene latex spheres (PSL) were nebulized and dried through a silica gel column (similar to the TSI 3062 type).The size selection was then refined with a Differential Mobility Analyzer (DMA). For particles larger than 350 nm, a Di-Ethyl-Hexyl-Sebacat (DEHS) solution was used to produce particles. After nebulization particles were dried and size selected with an aerodynamic aerosol classifier (AAC, Cambustion, UK). The aerodynamic diameter was later converted to mobility diameter for a more coherent comparison with the small particles selected with a DMA. A reference condensation particle counter (CPC) measured the particle number concentration after the DMA and AAC, while two CPCs were placed after the inlet. To represent the different tubing lengths inside the payload, one CPC was placed behind a short piece of black tubing (10 cm) and one was placed behind a longer piece (45 cm). The total flow through the main inlet was*

*1.72 lpm. Before the experiment, all CPCs were connected in parallel for direct comparison. Results from the CPC intercomparison are presented in Sect. 3.2.*

Section 3.2

RC44: Line 205: be precise, "particle number concentration", there are multiple other types of particle concentrations (surface, mass, …)

AR44: It has been corrected throughout the manuscript.

RC45: Line 206: provide the measurement range of the reference CPC.

AR45: The measurement range of the model 1720 is the same as for model 9403 (7 to 2000 nm). We included the information in the sentence. It now reads as follows:

*Two aMCPCs have been compared against a reference MCPC with the same measurement range (MCPC model 1720, Brechtel Manufacturing Inc., USA) with PSLs of $D_P$ 150 nm.*

RC46: Line 208: the figure should be part of the main manuscript

AR46: See answer below.

RC47: Line 210 to 214: This calibration is a fundamental part of ensuring validated measurements. Detailed information about test particles, averaging/comparison time, experimental setup (flow and line lengths), and the resulting plot should be part of the main manuscript. Please explain why there are two measurements for only some particle sizes and not for all. If there were multiple calibration runs, what explains the differences between the two runs, and why was one not done for the entire size range?

AR47: We have incorporated a comprehensive description of the experimental details in the main text.

However, given the extensive nature of the manuscript, we prefer to keep the figure in the SI to maintain the manuscript's focus. The same choice was made for Fig. S5. The characterization in question is essentially a validation of the manufacturer-provided calibration. Moreover, while the verification of the CPC activation diameter is important, the measurements and analysis performed with the MoMuCAMS are unaffected by it (provided that it is within a few nanometers from the nominal value). Hence, we believe the SI is the appropriate location for such details.

Regarding the duplicate measurements in Fig. S5, this is not indicative of multiple calibration runs. A single calibration run was conducted. The repetition of two points is attributable to the automated diameter scanning sequence, which recommended before termination, resulting in this duplication. We decided to keep these two separate measurement points to ensure transparency about the experiment and reproducibility of the measurement. We added a note about it in the manuscript:

*Note that the automatic scanning sequence produced two measurements for 7 and 7.5 nm particles. For transparency, results of both measurements are shown separately on Fig. S5.*

Section 3.3.

Line 224 to 227: Please provide more information on the size calibration:

- RC48: Calibration setup incl. dryer etc., average particle number concentration, sampling time, number of channels/size resolution of the mSEMS à was it the same as for balloon flights

  AR48: The information has been added to the manuscript. The revised text reads as follows:

*The performance of the mSEMS was tested with different particles covering its size range. Particles smaller than 50 nm were obtained by nebulizing pure MilliQ water using a portable aerosol generation system (PAGS, Handix scientific, USA). After nebulization, particles were dried through a silica gel dryer and size selected with a DMA. Particles larger than 50 nm were obtained by nebulizing PSL solutions and following the same procedure as with the pure MilliQ. For each size, particles were nebulized for over 10 minutes to allow enough scans to be counted. The mSEMS was set to 60 bins at 1 second per bin. The mobility diameter ($D_{mob}$) was obtained by fitting a lognormal distribution to the measured PNSD and taking the peak value (mean). Results of the experiments are presented in Fig. 4.a and Table 2. Overall, deviation in particle sizing, i.e. the relative difference between the particle size ($D_P$) and the measured distribution peak ($D_{mob}$) is below 7%.*

- RC49: Why wasn't the full measurement range from 8 to 300 nm calibrated?

  AR49: The initial provided sizing results were obtained from two experiments with the available PSL. We understand that providing results for the full size range is important. Using data from size selected particles from pure MilliQ during the first experiment, we were able to provide additional sizing results down to 8 nm.

  Moreover, a longer intercomparison between the mSEMS and SEMS is now provided in Sect. 3.5 (Figure 6 – see also answer to RC3 from reviewer 2)

  We believe that the sizing results and mSEMS – SEMS intercomparison provide sufficient evidence that the mSEMS is providing reliable measurements.

  RC50: It appears that the sizing uncertainty increased with decreasing particle diameters. A calibration of below 50 nm seems, therefore necessary

  AR50: The axis in Fig. 4 are in a log scale. Therefore, the error bars appear larger despite the standard deviation being typically less than ten percent of the particle diameter. Considering that there is no significant trend in the absolute difference between the measured and nominal particle diameter at different sizes we consider the results of these experiments to be satisfactory in showing that the mSEMS can reliably measure the aerosol size distribution. However, as mentioned above, sizing results down to 8 nm are now provided in Fig. 4 as well.

RC51: Line 227: seems inaccurate, the maximum deviation from the PSL diameter, including standard deviation, is larger than 8 %.

AR51: This is correct. We actually report the deviation between the measured peak diameter and the PSL (or MilliQ impurity) mean size. The measured width of the particle size is a combination of the mSEMS diffusional broadening and of the inherent transfer function of the first DMA. Hence, we consider the deviation between the measured peak diameter and the PSL mean size to be the best indicator for the mSEMS sizing accuracy. The distribution fit code has been refined and results indicate that the larger deviation is below 7% actually.

The information has been clarified in the manuscript.

*Overall, deviation in particle sizing, i.e. the relative difference between the particle size ($D_P$) and the measured distribution peak ($D_{mob}$) is below 7%.*

Detailed information on the measured particle size, standard deviation of lognormal distribution and deviation between measured peak diameter and the assumed real size are provided in Table 2.

RC52: Line 229: Was a dryer used after the nebulizer?

AR52: Yes, there was. It is now stipulated in the text.

RC53: Line 246-247: Is the explanation for the remaining underestimation of about 21% after correction supported by the number size distribution? The contour plots from the case studies don't seem to show a pronounced accumulation mode above 300 nm, which represents 21 % of the total particle number.

AR53: We believe that particles with sizes out of the 8-280 nm range (operating range of the mSEMS in this case) could indeed explain this difference. However, contribution from particles smaller than 8 nm are probably contributing to a large extend to the observed difference in this case, given that the standalone CPC has a $d_{50}$ cutoff of 5.7 nm. This explanation is supported by the measured PNSD during the comparison as shown in Fig. 1 (here below) where we can see an increase of very small nucleation mode particles.

[Figure]

*Figure 1: PNSD measured by the mSEMS during the intercomparison period of the mSEMS and aMCPC. The full line indicates the median. Grey shading represents the interquartile range.*

RC54: Line 253: what do you mean by "stratified layers" ?

AR54: We realize that the term "stratified layers" is ambiguous. We originally meant the presence of atmospheric layers due to a stratification of the atmosphere.

The sentence has therefore been changed to:

*Distance between each step varies according to the maximum altitude of the profile, desired time of flight and atmospheric conditions such as temperature inversions or stratification of the atmosphere.*

RC55: Line 250 to 255: This part rather reflects sampling strategy than instrument methods. Because it also appears in other sections, you should consider one separate paragraph for "sampling strategy".

AR55: We agree that the mentioned paragraph reflects more of a sampling strategy and should be presented separately. Following the reviewer's comment, we added a new section (4.1) to discuss various aspects of sampling strategies and flight operations. The specific paragraph (L250-255) has been moved to the new section.

Section 3.4

RC56: This whole paragraph needs methodological revision:

A deviation of 20 % at 500nm PSL seems high compared to the findings by Mei 2020, Liu 2021, and Pilz 2022. According to Gao 2016, Mei 2020 and Liu 2021, there should be no Mie resonances at 500nm with the POPS.

AR56: We took good note of the insights on the POPS characterization. However, the presented results are valid. The deviation between the PSL size and measurements remained within 10% for PSLs up to 800 nm for the POPS used for flights (POPS 105) and we believe that such deviation is acceptable given the uncertainties associated with optical measurements of particle size. We agree that Mie resonance does not occur below 600 nm (we also do not claim this in the manuscript text), which is well presented in Gao et al. (2016). However, for the PSL of 800 and 994 nm, Mie resonance becomes very evident as shown in Fig. 2 from Gao et al. (2016). Pilz et al. (2022) also reported higher deviations in the size range 500 to 1000 nm. The high deviation at 500 nm for POPS101, might be an instrumental issue. We find it important to mention here that this can occur with such type of instrument.

[Figure]

*Figure 2: Measured and theoretical optical signals calculated using Mie theory for the POPS. Figure from Gao et al. (2016).*

RC57: It appears that the instrument was rather not well adjusted or the optics were dirty, resulting in reduced signal intensities. Hence, all subsequent measurements should be taken carefully because all particles detected are actually larger than those measured by the POPS.

AR57: Although dirty or misaligned optics could explain part of the deviation, both units were characterized very shortly after purchase. The hypothesis of dirty optics seems therefore unlikely as the principal source of deviation in that case.

RC58: This has also an impact on the high noise that you observed in channels below 186 nm. It would mean that the noise of your POPS units is actually up to actual particle sizes of ca. 220 nm,

which would be very high compared to previous findings (Gao, Mei, Creamean, Liu, Pilz). In addition, dirty optics can also cause additional straylight leading to a reduced signal-to-noise ratio.

AR58: As shown on Fig. S7, the agreement between the two POPS and the CPC is very good when the first three bins are excluded from the analysis. This is a clear indication that the instrumental noise is not affecting particles larger than ca. 186 nm as indicated in the manuscript.

RC59: The fringes on the Gaussian signal you describe are not a regular behavior of the sensor and are not a sufficient explanation to oppose the previous findings. Your findings significantly reduce the useful size range of the POPS, particularly for your intended measurements in polar regions.

AR59: Although it had not been reported in previous literature, the explanation we suggest here comes from exchanges we had with the manufacturer and could explain why the phenomenon only occurs when particles are being measured. Here is the answer from Handix to our question:

*"The behavior you are describing is usually caused by "bumpy peaks". Instead of a Gaussian sticking up above a smooth baseline signal, there are little fringes on the leading and trailing edge of the peaks. The software picks those up as real, but small particles. When the filter is attached there are no big particles, and no little particles, so you see clean signal. When you put big PSL you get the little PSL."*

Given that the exact explanation is not entirely clear, we modified that text as follows.

*Results from Fig. S7 indicate that particles with diameters between 142 and 186 (bins 1 to 3) are wrongly detected by the POPS as total particle concentration increases. This phenomenon, potentially associated to stray light in the optics chamber, was already reported in previous literature (Gao et al., 2016; Mei et al., 2020; Pilz et al., 2022). According to the manufacturer, these wrong detections could also be explained by electronic noise from the detector, where fringes on the edge of the Gaussian signal are perceived as smaller particles by the software.*

Following RC4 from reviewer 2, the details of the characterization, initially placed in the SI were moved to the main manuscript.

Section 3.5

RC60: This whole section provides rather an experience report than an instrument characterization. There is no comparison to other instruments or calibration provided. The investigated impact of environmental conditions on measurements does not say much without a reference, and Figure 6 a) only shows a point cloud. What is the temperature in 6 b) ambient or inside the Pelicase?

AR60: We agree that the current presentation of the Pico represents a validation of its performance in flight in comparison to ground measurements. We mainly demonstrate that flight operations do not affect the instrument's baseline and therefore its functioning. We have therefore adapted the text to be more accurate.

*Only a few studies have provided information on the performance of the Pico instrument, however only for the methane ($CH_4$) version (Commane et al., 2022; Travis et al., 2020). This study provides a first experience of in flight operations of the CO version. [...]*

*[...] Although we demonstrate that vertical profiling does not affect the instrument's functionality, no quantitative characterization of the Pico's performance is available besides the manufacturer's calibrations. A comparison with a reference instrument or calibration gas should be done for future quantitative assessments of CO with the Pico.*

Former Fig. 6a (now 7a) shows the timeseries of the instruments baseline. The vertical spread of the blue dots indicates that flights experience similar variability as ground measurements. The temperature in former Fig. 6b (now 7b) represents the temperature inside the MoMuCAMS enclosure as indicated in the text.

To clarify this part, the text has been reworked:

*To identify, whether pressure or temperature changes have any influence on the instrument's baseline, several flights were performed in differential mode. Figure 7a shows the baseline measurement for a full campaign with color codes indicating whether the instrument was operated on the ground or in the air. Orange dots indicate that the instrument was operated inside a hut at constant temperature of about 20° C, while blue dots are baseline measurements when the Pico was inside MoMuCAMS in flight. Figure 6b shows in more detail the baseline variability on January 30, before, during and after a flight. The recorded inner temperature of MoMuCAMS and atmospheric pressure are indicated to illustrate the lack of correlation between changing environmental conditions and the instrument's baseline.*

RC61: What about manufacturer calibration or other publications that show the performance of the instrument?

AR61: Currently, we only found publications on the methane version of the MIRA Pico. Figure 3 has been taken from the instrument's brochure to illustrate the sub-ppb accuracy.

[Figure]

*Figure 3: Pico CO concentration stability measured from a tank with periodic rezeroing via the built-in scrubber. In this case, neither calibration nor zero gases are required to obtain 1ppb level accuracy for an indefinite period of time.*

Section 3.6

RC62: Line 313-314: What do you mean by this sentence: "The multiple nozzle-pattern achieves cut-size selection similarly to the more common Micro-Orifice Uniform-Deposit Impactors (MOUDI)".

AR62: The sentence at line 313-314 indicates that each stage is composed of multiple nozzles that achieve a size selection similarly to the more commonly known MOUDI. We have reworked the sentence, which now reads as follows:

*Each stage is composed of multiple nozzles, achieving size selection similar to the more common Micro-Orifice Uniform-Deposit Impactors (MOUDI).*

RC63: Is the HFI additionally equipped with a MOUDI, or is it compared to the MOUDI? Why is MOUDI written in brackets behind the instrument in Table 1?

AR63: The HFI is not additionally equipped with a MOUDI. We used the MOUDI acronym for its similarity with the HFI. However, we understand the confusion it might create and have removed the word "MOUDI" from Table 1.

RC64: Line 325 to 343: This part does not directly belong to the platform and should be considered as supplementary. The MoMuCAMS system provides a filter sampler, but it depends on the scientific objective and the environment of how the filters are treated and analyzed. Please keep in mind that this is a technical paper about a balloon platform.

AR64: We agree that this part is actually independent from the physical system and has therefore been moved to the SI. The initial Figure 10 (showing example of SEM/EDX analysis performed on collected particles) has also been moved to the SI.

Section 3.7

RC65: Line 362 to 379: Same as above.

AR65: The paragraph has been moved to the SI.

Section 3.8

RC65: Table 3: Accuracy is usually given as +-… and please use "to" instead of "-" for the ranges to avoid confusion with positive and negative values

AR65: Corrected.

RC66: Line 398 to 400: This is an outlook for future improvements and should be in the according section

AR66: The text has been modified. Because we already implemented those changes and tested the SHT85 sensor during the second experiment, we added the data to former Fig. 8 (now Fig. 9) and modified the text to mention the use of a new set of temperature sensors in addition to the SmartThether, which remains useful for wind and pressure measurements and T and RH in low radiation conditions.

This specific line has been moved to the end of the section and reads as follows:

*To address this issue, a solution including two sensors (SHT85, Sensirion, CH) in a shielding tube with active flow has been added to provide additional redundant T and RH measurements. Figure S1 in the supplementary material shows the new radiation shield on the MoMuCAMS box.*

RC67: Line 390: Please provide more details on how the comparison was done. Was the sensor on the tether or on the ground, distance to the reference station, etc.

AR67: During the first experiment, the SmartTether was directly attached on the tripod at the same height as the temperature sensor. During the second experiment, the snow cover was limiting our access to the tripod and the SmartTether was therefore placed on a structure holding it elevated at 50 cm above the snow layer and at about 1 meter from the tripod. The information has been added in the text.

*Two comparisons were performed on the ground between the SmartTether and a weather station equipped with a HygroVUE10 (Campbell Scientific) sensor, using an SHT35 sensing element (SHT35, Sensirion, CH). The first comparison was performed in Brigerbad, Switzerland on October 14, 2021. The second comparison was done in Fairbanks, Alaska on February 24, 2022. During the first experiment, the SmartTether was attached to the tripod of the weather station at a height of 2 m (same height as the reference temperature sensor). During the second experiment, the SmartTether was attached to a small structure at 50 cm above the snow and about 1 m from the tripod because of restrained access to the tripod due to important snow depth.*

RC68: Why was the sensor not shielded against direct radiation during comparison? It is common knowledge that temperature sensors are affected by direct solar radiation and are, therefore, always shielded in a weather station.

AR68: We fully agree that a temperature sensor should indeed be shielded. It is however the design of this specific sensor (also used by the atmospheric radiation measurement (ARM) research facility on their tethered balloons - https://www.arm.gov/publications/tech_reports/handbooks/tbs_handbook.pdf) and the purpose was to compare it in its original state. Due to the lack of the shield, we developed the new set of shielded temperature sensors that has been added to MoMuCAMS since then. We already had one SHT85 sensor during the second comparison, and data are now also shown in Fig. 9.

Section 4

This section can be shortened to focus on the platform's performance. The description of the single case partly suffers from a lack of ABL meteorology knowledge. I suggest focusing more on instrument/platform performance than on atmospheric science, which can only be partly done with the available ground-based and balloon-borne observations.

Please add:

- RC69: Internal temperature of the platform and relative humidity of sampled air during flights

  AR69: Profile examples at low temperatures and high relative humidity are shown in Fig. S2 (supplementary material) to illustrate the capabilities of MoMuCAMS. For the case studies, we always control that the RH is indeed below 40% but we do not show it for each flight to keep focus on the scientific description.

- RC70: Direct comparison of the mSEMS and the POPS with the ground SEMS in PNSD plots including discussion

  AR70: A figure (Fig. 6) showing a longer intercomparison between the SEMS, mSEMS and POPS has been added to section 3.5 with a discussion. Generally, all three instruments agree well.

  RC71: Performance evaluation of the mSEMS in terms of scanning time vs. accuracy, up-scan vs. down-scan, counting statistics in elevated atmospheric layers with low particle number concentrations including implications for measurements in polar regions.

AR71: The performance of the mSEMS in "up" and "down" scans shows no significant difference. Plumbing delay time is checked every time the mSEMS is disassembled and reassembled. We have added this information in sect. 3.3 and added a comparison figure in the SI.

*Comparison of "up" versus "down" scan performance of the mSEMS has shown no significant difference between the two modes. Results of a 6-hour averaged PNSD for up and down scans is shown in Fig. S8.*

The mSEMS is equivalent to any other type of scanning mobility particle sizer, and, as we have already shown, it can reliably measure the PNSD in its size range. Therefore, no instrument-specific consideration regarding scanning time versus accuracy and counting statistics is provided. Nonetheless, it is true that it may be necessary to increase the scanning time when measuring in clean environments with a low particle number concentration, which future tests and additional experiences will help to define. We included a mention of this in Sect. 3.3 and also in Sect. 4.1.

*(Sect. 3.3) The instrument is operated at a 0.36 lpm sample flow and 2.5 lpm sheath flow. We typically select a size range from 8 to 280 nm with 60 bins and a scan time of 1 minute (up scan) in regions close to anthropogenic sources. Note that the given values may need to be adjusted for environments with very low particle number concentrations (i.e. $< 100\ cm^{-3}$) to ensure good counting statistics, similarly to any electrical mobility sizer.*

*(Sect. 4.1) In the configuration described in Sect. 3.3, the mSEMS has therefore a vertical resolution between 13 and 20 m. For conditions with low particle number concentrations, the scan time might need to be increased to improve counting statistics, reducing therefore the spatial resolution. Users will need to define the best combination of bin time and number of bins (size resolution) to optimize the data quality and spatial resolution of the mSEMS.*

RC72: Performance of the inlet inside clouds, e.g. were water droplets entering the inlet?

AR72: The RH reduction of the inlet is demonstrated in Fig. S2. In terms of water droplets entering the inlet, we do not have any specific data besides the visual inspection after flight showing no signs of water intrusion into the inlet.

- RC73: Details on how optical PNSD from the POPS is merged with the mobility PNSD from the mSEMS

  AR73: We indicated in Sect. 3.5 that no conversion of the optical diameter from the POPS into the electrical mobility diameter was made. At this stage, the data processing and assumptions made to convert the optical PNSD to electrical mobility PNSD is considered out of the scope of this paper.

- RC74: Provide information about balloon climb rates and resulting vertical resolution for individual instruments

  AR74: Done. This information has been added to Sect. 4.1:

  *In this study, the velocity of the tether extension is 20 m per minute. The ascent and descent rate of the helikite depends on the line angle but based on discussion from Sect. 2.2, can vary between 13 and 20 m per minute for a zenith angle of 50 and 0°, respectively. The spatial resolution for instruments recording at 1 Hz is therefore between 0.2 and 0.3 m. In the configuration described in Sect. 3.3, the mSEMS has therefore a vertical resolution between 13 and 20 m.*

RC75: Line 412: What is a SEMS and what does it measure, mobility diameter? Please be precise with the abbreviation of particle number size distribution, either use PNSD or NSD. PSD could also mean volume or surface size distribution

AR75: The SEMS is the original version of the mSEMS. It stands for Scanning electrical mobility spectrometer. The acronym is now introduced in Sect. 3.5 and "PSD" has been corrected to "PNSD" throughout the text.

Section 4.1

RC76: Aerosol stratification and mixing usually result from ABL meteorology (besides aerosol direct radiative effects causing thermal layering in polluted environments). Therefore, it is useful to define ABL structure based on meteorological observations and derive aerosol layers from that. The authors should consider the book "An Introduction to Boundary Layer Meteorology" by Roland B. Stull

AR76: We note the comment and fully agree with it.

RC77: Line 434: Table 4 is not needed. Name the instruments of which measurements are shown.

AR77: Table 4 has been removed. The instrumental combination presented in each case study is now listed at the beginning with a short remark a scientific associated to the setup as requested in RC32.

RC78: Line 440: Figures 10 a) to c) should be considered supplementary since they are not related to the platform.

AR78: Figure 10 a to c has been moved to the SI. Only the original panel d was kept.

RC79: Figure 11: potential temperature should be used in the context of atmospheric stratification

AR79: The potential temperature has been added to panel a in dashed lines. However, for the identification of temperature inversion, the temperature is more convenient.

RC80: Please also add a humidity profile, this helps for a more accurate layer definition.

AR80: The relative humidity or specific humidity profiles do not exhibit a strong layering. Since they do not help with the interpretation, we did not add humidity to the figure as it is not discussed in the text.

RC81: Line 450: introduce N7-186 and be consistent with N>186 or N186 or N186-3370,

AC81: It has been introduced and corrected throughout the manuscript.

RC82: Why is the POPS range sometimes given up to 3000nm and sometimes up to 3370nm?

AR82: Thank you for pointing out this inconsistency. 3000 nm was used to round up the upper limit of the POPS. However, to remain consistent throughout the paper, we corrected this value t0 3370 nm.

RC83: Line 469 to 472: This doesn't seem reasonable. Wind shear is induced between layers of different wind directions or at the surface. Here, a typical logarithmic wind profile was observed that indicates wind shear at the surface with a change in wind direction (see: Ekmann layer).

AR82: Although wind shear also occurs with wind velocity difference, we agree that the given explanation is probably not accurate in this case. After careful analysis of the situation, it appears that

mechanical turbulences from wind shear near the surface causes mixing near the surface, which induced a mixed stable layer up to 50 meters. Mechanical turbulences then decrease progressively between 100 and 150 m leading to a negative gradient in pollutants. The sentence has been modified and reads as follows:

*This negative gradient can be explained by a progressive reduction of the turbulent mixing caused by wind shear at the surface, with altitude.*

RC84: The term "decouple" is usually referred to two turbulent atmospheric layers being disconnected by a stable layer. Here, mixing in a shallow surface layer inside the temperature inversion is induced by wind shear in the absence of solar radiation. The layers above are probably stably stratified, hence decoupling from an elevated mixed layer seems not the case.

AR84: Here the term decoupled refers to the decoupling between the surface layer and the residual layer above. This term is commonly used to describe the stable boundary layer (Mahrt, 1999). In the absence of mixing between the surface and residual layers, they are considered to be decoupled.

RC85: Line 477 to 479: The convection causes turbulent mixing, which leads to a decrease in stability, not vice versa.

AR85: We take good note of the comment and modified the sentence to:

*However, as the surface temperature increases between the first and last profile, convective mixing is induced and air from the residual layer is entrained into the surface layer.*

RC86: Line 486 to 490: This statement does not fit well. It appears very unlikely that the convection is strong enough to transport particles into the free troposphere from the bottom of the valley surrounded by ca. 1500 m peaks during fall and at the observed ABL conditions.

AR86: After consideration, the statement has been removed from the case study description, because there are no measurements to clarify the actual vertical reach of the surface pollution. We believe however that transport of surface emissions to elevated layers and potentially all the way to the free troposphere constitutes an interesting discussion point. We therefore address it in the conclusion to discuss the relevance of tethered balloon measurements with a combination of additional observations.

RC87: The phenomenon described by Harnisch could theoretically explain the observed RL during the first profiles. The presented observations do not seem sufficient to exclude this process.

AR87: We take note of the comment but as mentioned, the presented observations are not sufficient to validate this hypothesis and we will therefore not discuss this further in the manuscript.

Section 4.2

RC88: The ABL meteorology of this case does not seem to be accurately described referring to the book "An Introduction to Boundary Layer Meteorology" by Roland B. Stull

AR89: We take note of the reviewers' comment and adapted the description of the case study based on the comments below and revised it in order to be coherent with boundary layer meteorology theory.

Figure 15:

- RC89: should display the entire PNSD that was measured since it was emphasized in the text that this is an advancement

AR89: We understand the argument, even though the important advancement lies in having information on the smaller end of the particle size distribution; here, given the proximity to anthropogenic sources, the PNSD is dominated by sub micrometer particles. The POPS measured very low particle numbers at larger sizes and, therefore, we decided not to show the extended PNSD for graphic purposes.

- RC90: the POPS and the mSEMS distributions should be shown in different colors due to the different diameters

  AR90: Given the amount of colors on the plot, the addition of 3 more colors would make the figure hard to read. We preferred therefore to keep the original version where the POPS is indicated by dashed lines.

- RC91: x–axis label should contain D opt

  AR91: Corrected.

RC92: Line 559: the ground-based PNSD is probably better suited to determine the background PNSD

AR92: The background PNSD can be measured both at the surface and in the residual layer. However, as the ground-based PNSD is exposed to close anthropogenic sources, the very high concentrations from rush hours are masking the background PNSD. We actually see the two modes appearing more clearly after the erosion of the surface layer on panel e.

RC93: Line 563: Please specify why only a part of the ultrafine particles remain in the ABL

AR93: The initial number of ultrafine particle will decrease by coagulation processes. Part of the particles will also be lost by dry deposition and/or be mixed further up into the free troposphere. We included a short description in the sentence:

*Particles that are not lost via coagulation or dry deposition remain in the boundary layer after the development of a ML and grow to a size of about 40 nm.*

Section 4.3

RC94: Line 571: How is the desired sampling altitude defined?

AR94: The desired altitude is typically defined based on prior knowledge of the atmospheric structure. Typically, the free troposphere or the above cloud environment can constitute environments of interest. In this case, we have removed "desired", since these two flights were performed as proof of concept during a situation where the extent of the mixed boundary layer was higher than the reach of the helikite tether.

RC95: Line 573 to 575: This is rather a part of the sampling strategy than a result

AR95: This point of discussion has been moved to Sect. 4.1 were sampling strategies are discussed.

RC96: Line 576: The tether length was stated as more than 800m, why was sampling performed in the mixed layer?

AC96: The tether is not more than 800 m. Therefore, the maximum reachable altitude with this set of winches is 800 m without any wind. With wind, the altitude above ground level will decrease because of the angle in the line. Given the vertical extent of the boundary layer on these sunny days, the balloon could not get above the mixed boundary layer.

RC97: Line 580: Please provide more details on the aerosol mass concentration calculation from the optical PNSD and discuss sources for the uncertainty of up to 100 % for flight 2

AR97: The mass concentration has been calculated by converting the PNSD to volume size distribution, and integrated to obtain to total volume concentration, which has in turn been converted to a mass concentration assuming a mean particle density of 1.6 g cm$^{-3}$ which is a reasonable value given the predominance of anthropogenic sources (Pitz et al., 2003). The value in brackets does not refer to uncertainty but to the variability of the measured concentration throughout the sampling time.

The details have been added to the text:

*An estimation of the aerosol mass concentration during sampling time was calculated from PNSD measurements from the POPS. The PNSD was converted to a volume size distribution and integrated over all size bins to obtain the total volume concentration. The volume concentration was then converted to a mass concentration, assuming a mean particle density of 1.6 g cm$^{-3}$, given the predominance of anthropogenic sources (Pitz et al., 2003).*

RC98: Line 586 to 594: This is rather background information and no result belonging to the platform

AR98: The paragraph has been removed from the manuscript as suggested to keep the focus on the platform's performance.

RC99: Line 600 to 604: The sample flow is actually high for a balloon-borne filter sampler. If the 5 h sampling time in a mixed layer with close-by anthropogenic aerosol sources was almost too low, how do you see the capabilities for sampling in polar regions with one order of magnitude lower aerosol concentrations?

AR99: The low flow is in reference to a high volume sampler that would typically draw a flow rate of 1000 liters per minute but we agree that the flow rate is actually high for a balloon based sampler. We understand the concern about sampling at very low concentrations in polar regions and pointed this out in the manuscript. The text has been modified to read as follows:

*"The main limiting factor is the small aerosol mass concentrations obtained for the flight samples, which resulted in this case from a rather short sampling time. Great care must thus be taken in future studies in terms of sampling strategy to ensure that the amount of collected material is sufficient for chemical analysis, especially in polar regions, where mass concentration is typically much lower."*

Flight operations in polar regions are less restricted by regulation and could therefore be easily extended in time and would only be limited by battery capacity. Because of the remote control of the flow rate, it would also be possible to interrupt sampling, bring the balloon down for battery replacement and resume sampling, extending the sampling time considerably.

RC100: Line 606 to 616: This is no result of this study

AR100: This part has been removed from the manuscript to keep focus on the platform as suggested.

Section 5

RC101: The section number is wrong

AR101: Corrected

RC102: This section should be shortened to summarize the important technical advances of the system to atmospheric measurements, and case studies can be briefly summarized. Keep the focus on the technical paper!

AR102: The conclusion section has been revised to focus on the technical presentation of the system as suggested.

RC103: Line 623: measurements of optical properties were not shown, rather a concentration of trace gases was measured than a composition

AR103: Composition has been changed to concentration.
As stated above, although we did not present a case study for all instruments, measurements of light absorption are a feature of the MoMuCAMS and we believe that the reader should be aware of the measurement possibilities of the system. It is therefore important to have that information listed there.

RC104: Line 626: every altitude is relevant to the Earth's radiative budget

AR104: We acknowledge that. The revised sentence reads as follows:

*This information allows us to better study the origin of aerosol particles, their physical and chemical transformation and transport at different altitudes in the lower troposphere.*

RC105: Line 627: high wind speed and cold conditions were actually not shown

AR105: Please refer to AR9.

RC106: Line 632: High data quality was not sufficiently demonstrated, instrument inter-comparisons are missing, and measurement uncertainties are not entirely investigated

AR106: The sentence has been modified to:

*Because MoMuCAMS uses several relatively new instruments, laboratory and field characterizations have been performed to demonstrate their ability to provide accurate measurements.*

RC107: Line 633: What about the separate inlet for the filter sampler?

AR107: The filter sampler for microscopic analysis (FILT) uses the same inlet as other instruments.

The HFI stage impactor uses a short (~ 30 cm) and vertical inlet made of a 25-mm inner diameter Tygon tubing. Given the very short residence time (<0.1 s) and tubing orientation we expect losses to be negligible and did not characterize them.

RC108: "… deviation below 5% from…" deviation of what?

AR108: We meant deviation of particle number concentration. This element has been added in the manuscript.

*Two portable aMCPCs showed deviation of particle number concentration below 5% from a reference MCPC.*

RC109: Line 637: "The manuscript provides a first empirical correction function. " for what?

AR109: The function corrects for diffusional losses of ultrafine particles in the mSEMS (neutralizer, DMA and tubing). The information has been clarified in the text.

*We characterized the aerosol transmission efficiency through the mSEMS (including neutralizer, DMA and tubing) and showed that it is important to correct the measured size distribution for losses of ultrafine particles. The manuscript provides a first empirical correction function that can be used for this purpose.*

RC110: Line 639: the used number of channels is not a finding of this study

AR110: The sentence has been removed from the text.

RC111: Line 642: The STAP was not shown at all

AR111: Please refer to AR4.

RC112: Line 649: Pallas campaign was not shown

AR112: Please refer to AR 8.

RC113: Line 670: The reliability of the measurements cannot be proven by three case studies. Long-term comparisons along field campaigns should be considered for reliability evaluations

AR113: We corrected the sentence by removing this part. It now reads as follows:

*The characterization presented here provides a reference for future studies performed with MoMuCAMS.*

RC114: Line 675: What is "high signal to noise data"?

AR114: The term high signal to noise data was used in comparison to remote sensing data that can typically have a lower signal to noise ratio. However, to avoid any confusion, the sentence has been reworked and now reads as follows:

*Overall, MoMuCAMS is an easily deployable tethered balloon system able to cope with high wind speeds and cold conditions and to fly inside clouds, providing valuable data on the vertical distribution of aerosol and trace gases in different weather conditions.*

RC115: Line 675 to 679: Other airborne platforms proved the same capabilities. Actually, it is probably required to move along with an air mass to observe processes rather than staying at one spot constantly taking snapshots

AR115: We agree that other platforms are better suited to perform a Lagrangian experiment, i.e. following the air mass. However, the helikite based system is probably one of the best to make observations with such a comprehensive payload at low altitudes within a moving air mass. Particularly in polar regions, measurements variations of trace gas mixing ratios and particle number concentrations indicate that air mass characteristics change on the order of hours, rather than shorter time scales. Hence hovering with the helikite over a few hours will allow us to study the same air

mass, despite it moving. Generally, drones, aircraft and other tethered balloon systems will also have difficulties following an air mass at the speed it travels. In addition, aircraft cannot fly at low altitudes (from the ground to a few hundred meters) in low visibility conditions, which results in an observational gaps. Other tethered-balloons and blimps are very limited by wind speed and drones are very susceptible to icy conditions and have very limited lifting capacity compared to a tethered balloon system.

The helikite has the capability of providing profiles and longer measurements at given altitudes that can be compared to the ground. Thus, a direct comparison between the airborne and ground measurements can provide insights on processes happening higher up. We recognize that all these different platforms are complementary and combination of these different observational techniques is needed to obtain a full picture of the atmospheric processes.

Based on the reviewer's comment, we have adapted to concluding paragraph as follows.

*Overall, MoMuCAMS is an easily deployable tethered balloon system able to cope with high wind speeds and cold conditions and to fly inside clouds, providing valuable in situ data in different weather conditions. Its ability to cope with harsh environmental conditions combined with the presented suite of instruments will contribute to providing new insights in the vertical distribution of aerosol and trace gases in the lower atmosphere.*

**Answers to Reviewer 2**

Anonymous referee #2, 11 May 2023

Summary and overall impression

This manuscript describes a new measurement platform that can be deployed on a balloon/kite to obtain aerosol and trace gas measurements from the surface to 500m above the surface (although I believe the tether can reach 800m). The platform uses several miniature instruments. The manuscript describes and characterizes the platform inlet and several (but not all) of the instruments and shows some field data collected using the platform. I recommend publication after minor revisions.

We thank the reviewer for their constructive comments.

Comments

RC1: Figure 3 a. Is this for large particles? The text suggests that the sampling efficiency is only impacted by particles > 2um.

AR1: Figure 3a shows the inlet sampling efficiency (y-axis) as a function of particle size (x-axis) for particles between 8 and 3000 nm, which is the size range measured by our system.

RC2: Why is there no discussion of the $CO_2$, $O_3$, and STAP instruments? Are there data for the $O_3$ and STAP instruments?

AR2: As indicated in line 159, with exception of the POPS, we present instruments that have not been characterized in previous studies with the aim to focus on novelties and keep the paper to a reasonable length. The STAP has been presented and well characterized by Pikridas et al. (2019) and Pilz et al. (2022) for similar applications. The model 205 from 2BTech is a commonly used instrument for ozone measurements and has been featured in many scientific publications (https://twobtech.com/citations.html). The Vaisala $CO_2$ sensor is also common and its performance in flight has been presented in Brus et al. (2021). The introduction of section 3 has been completed to clearly state that information and provide all references for the reader.

*In this section, we provide a detailed characterization of the inlet system (Sect. 3.1), and present instruments used on MoMuCAMS, which have not already been described in previous publications. In particular, we present the advanced mixing condensation particle counter (aMCPC) (Sect. 3.2, see Table 1 for abbreviation), miniaturized scanning electrical mobility sizer (mSEMS) (Sect. 3.3) and Mira Pico gas analyzer (Sect. 3.5). The printed optical particle spectrometer (POPS) was described already by Gao et al. (2016) and Mei et al. (2020); nonetheless, we present here a characterization of our POPS (Sect. 3.4) because it constitutes a reference instrument on the MoMuCAMS. Additionally, setups for filter based sample collection for chemical composition analysis and electron microscopy are described in Sect. 3.6 and 3.7, respectively. Performance of a meteorological sensor (SmartTether, Anasphere, USA) is presented in Sect. 3.8. The reader is referred to Pikridas et al. (2019) and Pilz et al. (2022) for a description of the STAP (model 9406, Brechtel Manufacturing Inc., USA). For the more commonly used ozone monitor (model 205, 2BTech, USA), the reader can refer to the Atmospheric Radiation Measurement (ARM) ozone handbook (Springston et al., 2020) and for an evaluation of flight performance of the carbon dioxide monitor (GMP343, Vaisala, Finland), the reader can refer to Brus et al., (2021).*

RC 3: How well do the POPS and mSEMS agree in the region of overlapping size (186-300 nm)?

AR3: To address this comment, a discussion on the comparison of the mSEMS, SEMS and POPS has been added to section 3.5. The three instruments sampled from the same inlet from January 30 to January 31, 2022. Figure 4 (6 in the manuscript) shows results of the intercomparison. On panel (b), we can see the comparison of the total particle number concentration for the POPS and the mSEMS for the same size range (186 to 1500 nm). The linear regression slop is 0.89.

The overlapping of the number size distributions of the POPS and mSEMS is shown on panel (c), and more specifically in the top right corner. A quantitative assessment of the size depended particle-counting shows a good agreement between the POPS and the mSEMS within 5%. Comparison of the POPS with the SEMS show slightly larger differences (up to 20%).

[Figure]

*Figure 4: Comparison of the mSEMS, SEMS and POPS between January 30 and 31, 2022. Measurements were performed at the University of Alaska farm field in Fairbanks, USA (64°51'12"N / 147°51'34" W). a) Timeseries of particle number concentration from 8 to 280 nm ($N_{8-280}$) from mSEMS (red) and SEMS (blue). b) Timeseries of particle number concentration from 180 to 1500 nm ($N_{180-1500}$) from POPS (green) and SEMS (blue). c) Particle number size distribution measured from 02:00 and 04:00 on January 31 (shaded grey area in (a) and (b).*

RC4: I think sections S1 and S2 belong in the main manuscript.

AR4: Details from the experiments (previously in the SI) and results have been added to the manuscript as suggested. Additional details on the experimental design for the aMCPC tests were added. However, given the extensive nature of the paper and that the figures essentially represent validation from the manufacturer's calibration or previously published work, we kept the figures in the SI.

**Answers to Reviewer 3**

Anonymous referee #3, 18 May 2023

Summary and overall impression

This article details MoMuCAMS, a new instrumentation system that can be deployed on tethered balloons or UAVs to collect vertical profiles of aerosols, meteorological parameters, trace gases, and microscopy samples. The article details the instrumentation that composes the MoMuCAMS and presents field results from three locations. Publication is recommended after minor revisions.

We thank the reviewer for their constructive comments, which helped in improving the quality of the manuscript.

Comments

RC1: Line 65, remove comma

AR1: Done.

RC2: Line 94:101, refer to these as instruments not tethered balloon systems or UAVs. The current reference to these as tethered balloon systems indicated that the author is referring to developments in flight platforms rather than the instrumentation packages.

AR2: We understand the confusion between instrument and flight platform development. After careful consideration, we believe that the term "instrumental platform" would be the most appropriate term as it refers directly to the instrumental package but includes the specific structure holding the instruments and running them. Additionally, following reviewer 1 comments, we shortened this part of the introduction which now reads as follows:

*Recently, there have been important developments in both UAV and tethered balloon instrumental platforms (Bates et al., 2013; Ferrero et al., 2016; Mazzola et al., 2016; Pilz et al., 2022; Porter et al., 2020). The platforms referenced above have typically been designed for specific targets and have therefore limited freedom in instrumental setup modification.*

RC3: Line 114, quantify windy. Aerostats are advertised to operate in hurricane force winds.

AR3: The manufacturer states that the 45 $m^3$ can fly in wind speeds up to 22 $ms^{-1}$. From our experience, the maximum wind speed we have measured during flights was slightly above 15 $ms^{-1}$ and the balloon was indeed stable and the instrumental package remained in its upright position with minimal sideways swinging motion. The manufacturer indication has been added in parenthesis in the main text.

RC4: Line 115, if you list balloon setup it should be better defined

AR4: The balloon setup includes the balloon itself, two main ground anchors, a landing inflatable helibase and 20 metal stakes for additional anchoring points.

Following recommendations from other reviewers, the introduction has been shortened to focus on the essential. Since the paper is about the technical aspect of MoMuCAMS, which is independent from the helikite itself, this part has been deleted from the introduction.

RC5: Line 135, since earlier in the paper you state that tethered balloons are not as bounded by UAVs in flight time, quantify the expected operating time and the Ah of the batteries

AR5: We use 6-cell Lipo batteries of different capacities ranging from 9 to 22 Ah for a nominal voltage of 22.2 V. The operation time will depend on the selected batteries, the instrumental setup and the air temperature and is complicated to summarize based on all these parameters. However, following the reviewer's comments, we added the following information in the text.

*Batteries with a capacity between 9 and 22 Ah and a nominal voltage of 22.2 V are typically used. The maximum flight operation time will depend on the selected batteries, instrumental setup and ambient temperature but usually ranges from 2 to 10 hours.*

RC6: Line 150, the line length is referenced as 800m but the highest mean altitude presented in Table 5 is 434m, which would indicate a steep tether angle. Since the authors promote the helikite as a tethered balloon system due to flight in stable winds, a discussion of the expected tether angle would be beneficial. In general, if the authors are claiming the helikite excels as a tethered balloon system,

the flight characteristics of the platform throughout the experiments should be detailed or these references in the publication should be removed.

AR6: It is true that the angle between the line and the zenith will increase with increasing wind speed and therefore, reduce the maximum altitude the balloon can reach.

To address the reviewer's comment, we investigated the altitude, horizontal drift and wind speed of flights for all flights performed during the Brigerbad, Switzerland and Pallas, Finland field campaigns to provide an approximation of the line angle in response to wind speeds.

The results are however only indicative and depend on the difference between balloon's lift and payload weight, payload positioning on the helikite etc. Note also that the manuscript shows results with a 45 m$^3$ helikite with an 800 m long tether. The MoMuCAMS system is however independent from this helikite and can be installed on a larger balloon with longer tether to reach higher altitudes. This has now been also stipulated in the manuscript.

This additional information has been added to Sect. 2.2

*As wind increases, the zenith angle of the line increases as well, reducing the maximum altitude reachable with the helikite. The angle depends on the wind speed but also the net lift, which will depend on the atmospheric pressure, inflation state of the balloon, presence of water, weight of the payload and tether. Estimates of zenith angles have been calculated from the horizontal displacement of the helikite (measured by GPS) and its altitude above ground level. Figure S2 (supplementary material) shows results for two fields campaigns. Generally, the zenith angle tends to stabilize between 45 to 50° at around 8 to 10 ms$^{-1}$. Note that the MoMuCAMS is independent of the helikite and a larger model with longer tether can be used to reach higher altitudes if needed.*

and the following figure was added to the SI.

[Figure]

*Figure 5: Box plots of the helikite's tether zenith angle against measured wind speed. The zenith angle was estimated from the horizontal displacement given by recorded GPS location during flight and the calculated barometric altitude. Colors are indicative of two field campaigns. Orange corresponds to*

*Brigerbad (46°18'00"N / 7°55'16" E), in a Swiss alpine valley and red corresponds to Pallas in Finland (68°00'00'' N / 24°14'22" E).*

It is also good to note that this issue concerns all types of tethered balloon. The difference of the helikite is that it will preserve its stability and still fly when other types of balloons are known to be pushed down by the wind and therefore cannot be operated at wind speeds above 10 or sometimes even 5 ms$^{-1}$.

RC7: Line 154, what location is referred to for these ranges?

AR7: The coldest temperature was measured in Fairbanks, USA and the highest wind speeds were measured in Pallas, Finland. Vertical profiles of these flights are shown in Fig. S2. The location has been indicated in the main text.

RC8: Line 172, prevents

AR8: Done.

RC9: Line 404, the authors should quantify low and high

AR9: Low refers to overcast conditions or at night. Sufficiently high wind speeds refers to wind speeds of at least 1 ms$^{-1}$, which allow the sensor to be remain in its horizontal position and therefore be shield from direct sun exposure. The information was added in the main text.

*Overall, the SmartTether provides reliable measurements when solar irradiance is low (overcast skies or at night) and/or wind speed is sufficiently high (> 1 ms$^{-1}$). In other cases, measurements can be biased and data should be treated accordingly.*

However, as indicated in the revised version, additional sensors have been added to address the radiation sensitivity issue of the SmartTether and temperature measurements are therefore no longer affected by sunlight.

RC10: Line 496, is 13d really the balloon altitude, or the sensor package? How is the altitude measured and what is the expected accuracy?

AR10: As the sensor package is directly attached to the helikite, we can consider that both altitudes are equivalent. The altitude is calculated using the barometric formula. To determine the uncertainty of the calculated altitude, we used measurements of the altitude at a fixed elevation above ground for 3 hours. Using the root mean square error (RMSE), we calculate that the uncertainty was 0.97 m, which we round up to +/- 1 m. The information has been added to Sect. 4.1.

*Altitude during flight is provided by the GPS of the SmartTether and is re-calculated during post processing of the data using the barometric formula (Eq. 3),*

$$h_b = \frac{T_0}{L_0}(1 - \frac{p_b}{p_0})^{\frac{L_0 R}{g}}$$

*(3)*

*where, $T_0$ is the temperature at the surface, $L_0$ = 6.5 K km$^{-1}$ is the mean environmental lapse rate, $p_0$ and $p_b$ is the pressure at the surface and balloon height, respectively, R = 287 J kg$^{-1}$ K$^{-1}$ is the gas constant for dry air and g is the Earth's gravitational acceleration. An uncertainty of ±1 m for the*

*altitude was calculated using the root mean square error for a 3-hour time series of altitude measurement at a known altitude.*

RC11: Line 519, this sentence should be reworked

AR11: The sentence has been reworked to improve readability:

*Finally, after 11:15 [P3], the surface layer is eroded and the entire vertical column looks more homogenous.*

RC12: Line 629, the authors frequently reference stability. How is this defined and where are these results presented? The authors also reference leakage from the internal membrane below -20 °C but refer to operating at -36 °C. It would be informative to better quantify the helium loss below -20 °C in comparison to higher temperatures if the authors frequently want to reference these statistics as advantageous to the helikite.

AR12: The stability is only assessed through visual observations during flight. The inlet always faces into the wind and remains horizontal. We have observed that even at high wind speeds, both the balloon and the payload remain very still and instruments that are sensitive to tilt have not experienced issues such as butanol flooding of CPC optics during flights. This information is provided as an indication that helikites are suitable for applications in cold and windy conditions.

Unfortunately, quantifying helium losses below -20°C is difficult. The helikite is composed of two layers. An outer shell provides the structure of the helikite, while an inner polyurethane membrane holds the helium. As temperature drops, the polyurethane becomes brittle and very small punctures appear. As indicated in former line 155, we estimate that the punctures appear at temperatures below -20°C as operations at temperatures above this threshold have typically not led to any major helium loss from our experience. However, given the size of the punctures and the fact that the membrane is inside the external shell, it is not possible to tell when the punctures occur and how fast helium is really being lost, as it is a slow process. We therefore can only share our best estimate for any future user who might be interested in using a helikite at such temperatures.

RC13: In general, since the authors frequently reference that this system can measure at lower particle diameters than other systems, this work seems to somewhat arbitrarily dismiss all data in the three lowest POPS size bins. More support for the exclusion of all three bins should be provided, especially since these bins typically represent the most commonly encountered particle sizes in most environments.

AR13: As indicated in section 3.4 and Fig. S7, bins 1 to 3 show increasing noise level with increasing particle concentrations, leading to overestimations of particle number concentration in the size range 142 to 186 nm of the POPS. Noise level were already reported in existing literature (Pilz et al., 2022) and Fig. S7 supports that total particle concentration agrees better with our reference CPC when bins 1 to 3 are removed. However, the mSEMS provides size distribution for particles down to 8 nm, which is in any case much lower than what an optical particle counter can detect. Note that the Aitken mode is actually the most prevalent in polar regions as reported by Boyer et al. (2023) and Schmale et al. (2019). The argument of the size distribution is therefore mainly based on the contribution of the mSEMS yielding a size distribution from 8 nm to 300 nm, completed by the POPS, which provides a size distribution up to 3370 nm.

RC14: All Figures, since the authors made observations at multiple locations it would be beneficial to

include the location in the plot or captions. Especially since the authors reference local emissions sources.

AR14: The location and GPS position has been specified in the caption of the figures. For the case studies, the location is only indicated in the first "flight" figure as it is also already indicated in the manuscript.

---

## Author Response (AR2)

We thank the editor and reviewers for the handling and commenting of our revised manuscript. Please find below our point-by-point responses to each of the reviewers' comments, including the modifications we have made to the manuscript. Editor comments (EC) and reviewer comments (RC) are in black text and the answers to reviewers (AR) are given in blue text, and excerpts of the revised manuscript are given in *blue italic text.*

**Answer to editor's comments**

EC1: L16: you here use the term "payload", but then subsequently describe MoMuCAMS as the container for instruments, which would seem to be the real payload.

AR1: That's correct. We changed the word payload to platform, which is a more generic term describing the enclosure with all the included systems.

EC2: L20: should "a board computer" —> "a single-board computer"?
AR2: Done

EC3: L22: m3 superscript

AR3: Corrected.

EC4: L23-25: something about how the flexibility allows for different instrument configurations to be used depending on the science focus of the observations?

AR4: The following sentence has been added at line 25:

*Different instrumental combinations are therefore possible to address the specific scientific focus of the observations.*

EC5: L28: "hitherto not yet characterizied" —> "previously unreported"?
AR5: The sentence has been changed as suggested.

*Here we present a characterization of the specifically developed inlet system and previously unreported instruments, most notably a miniaturized scanning electrical mobility spectrometer and a near-infrared carbon monoxide monitor.*

EC6: L31: "the feasibility of" could be omitted
AR6: It has been removed.

EC7: L39: "allows to observe" —> "allows observations of"
AR7: Corrected.

EC8: L43: maybe "key challenges in atmospheric science" since there is a realm realm of aerosol science that is unrelated to the atmosphere

AR8: The sentence has been rephrased as suggested.

*One of the key challenges in atmospheric science is understanding the large heterogeneity of aerosol particles in space and time.*

EC9: L56: "Poles" —> "Polar regions and alpine valleys are two environments where"
AR9: Done.

EC10: L106: "flies" —> "carries"?

AR10. Indeed, this sounds better. It has been corrected.

EC11: L131: "is" —> "was" or "has been" since you are referring to the deployments so far, but other platforms could be used

AR11: Done.

EC12: L160: "STAP" —> "Single-channel Tricolor Absorption Photometer (STAP; model 9406…"
AR12: Done.

EC13: S3.4: choice of refractive index (not stated/discussed) can make a difference in the analysis you have conducted. Calibration was performed with PSLs which have a RI ~ 1.61, but tropospheric particles are likely to be lower (more in the 1.50 – 1.55 range (real component) for org-sulf particles, have an imaginary component (e.g. for biomass burning/soot), and be quite different for mineral/salt (as well as possibly not spherical)), which affects sizing and consequently the measured number given the impact on the low cut size and the steepness of the size distribution in that size range.

AR13: We agree that the aerosol refractive index and shape can have a large effect on the sizing accuracy of the POPS (and most other OPCs). However, it is out of the scope of this work to characterize this effect, especially considering that it has already been investigated by previous studies (see for example Mei et al., 2020). As correctly pointed out by the editor, the representative refractive index for tropospheric particles is in the 1.50 – 1.55 range which is close enough to that of PSL to only have a minor effect on the sizing accuracy of the POPS. This was also shown in the Mei et al., 2020 study mentioned above. Nevertheless, we acknowledge that in environments with aerosols having markedly different optical properties – for example, in arid regions with a high concentration of mineral dust – the data could be significantly affected and should be treated accordingly. We added a comment for the reader at line 282:

*Note that the sizing characterization was performed with PSLs with a refractive index of 1.59. The refractive index of tropospheric aerosol particles typically is in the 1.50 – 1.55 range which is close enough to that of PSL to only have a minor effect on the sizing accuracy of the POPS. However, if measurements were to be conducted in environments with aerosols having markedly different optical properties – for example, in arid regions with a high concentration of mineral dust – the data could be significantly affected and should be treated accordingly.*

EC14: L280: describe spacing of size bins, e.g. log spaced?

AR14: The bins are indeed log spaced. The information has been simply added in the sentence at line 281-282.

*We follow therefore their recommendations by setting the POPS size resolution to 16 log spaced bins to minimize sizing errors.*

EC15: L287: "particles with diameters between 142 and 186 (bins 1 to 3) are wrongly detected" —> "the measurement of particles with diameters less than 186 nm (bins 1 to 3) are affected by measurement artifacts that result in inflated apparent particle counts that scaled with particle concentration."

AR15: The sentence has been corrected as suggested.

EC16: L296: "in parallel of" —> "in parallel with"

AR16: Done.

EC17: L301: Might clarify/restate that the SEMS – POPS comparison shown in Fig 6b (y-axis indicates Dp = 180-1500 nm) is using a slightly different lower POPS diameter > 186 nm (bins 4-16—correct?) And as you have noted, the comparison is also mobility diameter vs optical diameter.

AR17: An indication about the slightly different size ranges was already present at line 305. We completed it by restating the fact that the difference in types of measured diameters and the exact size range of the POPS.

*Note that the size range of each instrument differed slightly because of respective bin limits and different types of measured diameters (i.e. mobility versus optical diameter / POPS size range for bins 4 to 14 = 186 to 1480 nm).*

EC18: L433: "are typically applied with MoMuCAMS" —> "have been utilized for sampling with MoMuCAMS"

AR18: Done.

EC19: L459: "p0 and pb is" —> "p0 and pb are"

AR19: Done.

EC20: L548: "surface layer (SL)"

AR20: (SL) has been added.

EC21: L566: "biased to surface" —> "biased the surface"?

AR21: Indeed. It has been corrected.

**Answers to reviewer #1 comments**

RC1: Line 56: „Poles" should be polar

AR1: The sentence has been corrected as follows:

*Polar regions and alpine valleys are two environments were a stable boundary layer is commonly observed.*

RC2: Lin 147: please add the resulting maximum height at the typical 45° inclination angle with the 800m tether

AR2: The height at the given angles has been added. The revised sentence reads as follows:

*Generally, the zenith angle tends to stabilize between 45 to 50° at around 8 to 10 ms$^{-1}$, which corresponds to a maximum altitude between 515 and 565 m a.g.l. for an 800 m long tether.*

RC3: Line 193: "height" probably has to be "eight"

AR3: Indeed, it has been corrected.

RC4: Line 194: the largest particle diameter shown in the plot is actually 3000 nm

AR4: That's true. It was a mistake in the text. It has been corrected to 3000 nm.

RC5: Line 222-223: The plateau efficiency (A) of both CPCs is not reached at 7 nm in plot S5! Please correct

AR5: We agree that the plateau is actually located more between 8 and 9 nm, although, following equation 1, values at 7 nm already indicate counting efficiencies around 100%. The text has been revised to:

*The detection efficiency for both aMCPCs reaches a plateau between roughly 8 and 9 nm, which is in agreement with the manufacturer's specifications.*